# Exploiting Activation Sparsity with Dense to Dynamic-k Mixture-of-Experts Conversion

**Filip Szatkowski**[*]
IDEAS NCBR
Warsaw University of Technology

**Bartosz Wójcik**[*†]
IDEAS NCBR
Jagiellonian University [‡]

**Mikołaj Piórczyński**[*]
Warsaw University of Technology

**Simone Scardapane**
Sapienza University of Rome

## Abstract

Transformer models can face practical limitations due to their high computational requirements. At the same time, such models exhibit significant activation sparsity, which can be leveraged to reduce the inference cost by converting parts of the network into equivalent Mixture-of-Experts (MoE) layers. Despite the crucial role played by activation sparsity, its impact on this process remains unexplored. We demonstrate that the efficiency of the conversion can be significantly enhanced by a proper regularization of the activation sparsity of the base model. Moreover, motivated by the high variance of the number of activated neurons for different inputs, we introduce a more effective dynamic-$k$ expert selection rule that adjusts the number of executed experts on a per-token basis. To achieve further savings, we extend this approach to multi-head attention projections. Finally, we develop an efficient implementation that translates these computational savings into actual wall-clock speedup. The proposed method, Dense to Dynamic-$k$ Mixture-of-Experts (D2DMoE), outperforms existing approaches on common NLP and vision tasks, reducing inference cost by up to 60% without significantly impacting performance.

## 1 Introduction

Transformers have become a predominant model architecture in various domains of deep learning such as machine translation [47], language modeling [6, 31], and computer vision [7, 21]. The widespread effectiveness of Transformer models in various applications is closely related to their ability to scale efficiently with the number of model parameters [20], prompting researchers to train progressively larger and larger models [45, 19]. However, the considerable computational demands of these models often restrict their deployment in practical settings with limited resources.

At the same time, Transformer models exhibit considerable activation sparsity in their intermediate representations [24], which suggests that most of their computations are redundant. Conditional computation methods can reduce these unnecessary costs by using only a subset of the model parameters for any given input [14]. In particular, Mixture-of-Experts (MoE) layers [38], consisting of multiple experts that are sparsely executed for any input token, are an effective way to decouple the number of parameters of the model from its computational cost [3]. As shown by [52], many pre-trained dense Transformer models can be made more efficient by converting their FFN blocks into MoE layers, a process they call *MoEfication*.

---

[*]Equal contribution
[†]Corresponding author: b.wojcik@doctoral.uj.edu.pl
[‡]Faculty of Mathematics and Computer Science, Doctoral School of Exact and Natural Sciences

38th Conference on Neural Information Processing Systems (NeurIPS 2024).

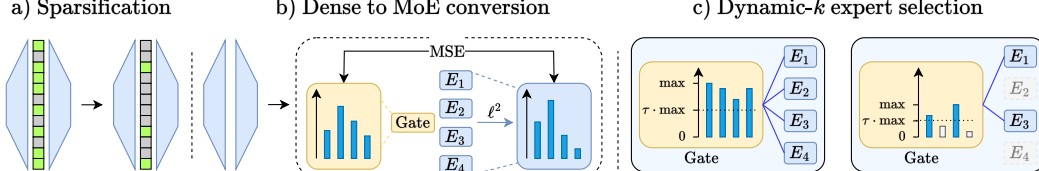

a) Sparsification    b) Dense to MoE conversion    c) Dynamic-$k$ expert selection

Figure 1: Key components of D2DMoE: (a) We enhance the activation sparsity in the base model. (b) We convert FFN layers in the model to MoE layers with routers that predict the contribution of each expert. (c) We introduce dynamic-$k$ routing that selects the experts for execution based on their predicted contribution.

**Contributions of this paper**: We consider the following research question: what is the *optimal* way to convert a generic Transformer model into an equivalent sparse variant? We identify a series of weaknesses of the MoEfication process limiting the resulting accuracy-sparsity tradeoff, and propose corresponding mitigations as follows. We call the resulting algorithm Dense to Dynamic-$k$ Mixture-of-Experts (D2DMoE) and outline it in Figure 1.

1. First, we analyze the relationship between the activation sparsity of the starting model and the efficiency of the final MoE model. We show that computational savings are directly related to sparsity levels, and we correspondingly enforce higher activation sparsity levels before conversion through a lightweight fine-tuning process, which leads to a substantially improved cost-to-performance trade-off.

2. We identify the router training scheme in the original MoEfication algorithm as a limitation of the conversion process. We propose to frame the router training as a regression problem instead, hence our routers directly predict the norm of the output of each expert.

3. We show that Transformer models exhibit significant variance of the number of activated neurons, and standard top-$k$ expert selection in the MoE layers is inefficient. We propose an alternative dynamic-$k$ expert selection scheme that adjusts the number of activated experts on a per-token basis. This approach enables the model to efficiently allocate compute between easy and hard inputs, increasing the overall efficiency.

4. We generalize the conversion method to any standalone linear layer including gated MLP variants commonly found in modern LLMs [45, 42] and projections in Multi-Head Attention (MHA) layers (which often account for over 30% of total computations in Transformer models [39]).

   For MHA, we propose a replacement procedure in which every dense layer is substituted by a two-layer MLP module trained to imitate the output of the original layer.

We evaluate D2DMoE across benchmarks in text classification, image classification, and language modeling, demonstrating significant improvements in cost-performance trade-offs in all cases. D2DMoE is particularly well-suited for contemporary hardware, as evidenced by our efficient GPU implementation, which we contribute alongside our proposed method.

## 2   Motivation

MoE models have gained a lot of traction over the last years as an effective architecture to decouple the parameter count from the computational cost of the models [56]. In a MoE layer, hard sparsity is usually enforced *explicitly* by applying a top-$k$ operation on the outputs of a trainable gating layer. However, many recent works [53, 2, 30] have shown that most Transformers, when trained at scale, build *intrinsically* sparse and modular representations. Zhang et al. [52] proposed to leverage this naturally emerging modularity with MoEfication - a method that converts dense transformer models into MoE models by grouping FFN weights into experts and subsequently learning small routers that determine which experts to activate. Models converted with MoEfication are able to preserve the performance of the original dense models while using only a fraction of their computational cost. However, we believe that the MoEfication procedure is not optimal, and therefore aim to obtain dense-to-sparse conversion schemes that obtain a better cost-performance trade-off.

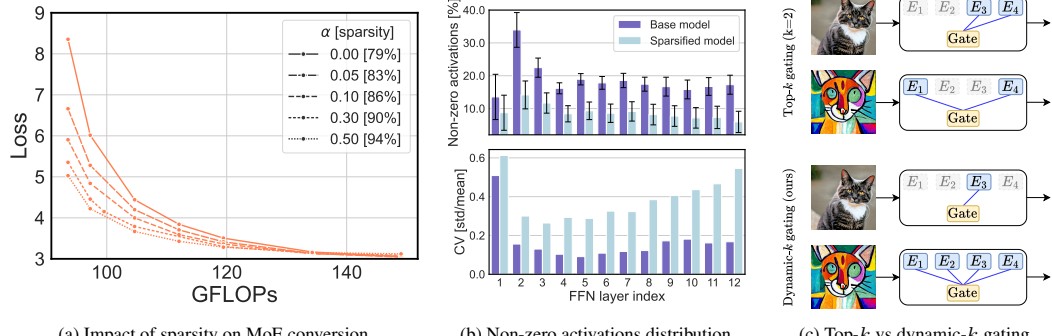

(a) Impact of sparsity on MoE conversion     (b) Non-zero activations distribution     (c) Top-$k$ vs dynamic-$k$ gating

Figure 2: (a) Cost-accuracy tradeoff for a *MoEfied* [27] GPT-2 model obtained starting from models with different levels of activation sparsity. Sparsification correlates with the model performance. (b) Distribution of non-zero activations in the FFN layers in GPT-2-base on OpenWebText, with and without the sparsity enforcement phase. Both models exhibit significant variance, and the mean-to-variance ratio increases in the sparsified model. (c) We propose to exploit the variation in activations through a dynamic-$k$ routing procedure that adapts the number of experts allocated to a sample.

Intuitively, a MoE converted from a sparser base model would be able to perform the original function using a smaller number of experts. To validate this hypothesis, we perform MoEfication on different variants of GPT2-base[4] with varying activation sparsity levels and show the results in Figure 2a. As expected, MoEfication performs better with sparser models. We further investigate the per-token mean and the variance of non-zero neurons in the base and sparsified model, and show the results in Figure 2b. Observe that different layers use a different number of neurons on average. Moreover, the variance of the number of activated neurons is quite high and becomes even more significant in the sparsified model. This means that static top-$k$ gating as used in MoEfication is not optimal for dense-to-MoE converted models, and a more flexible expert assignment rule that would be able to handle the high per-token and per-layer variance could be beneficial to the efficiency of such models, as illustrated at Figure 2c. Such dynamic-k gating requires routers that reliably predict the contribution of each expert. We observe that routers obtained through MoEfication do not accurately reflect this contribution. Moreover, their router training procedure depends on the strict sparsity of the model guaranteed by the ReLU activation function. Therefore, we design a novel router training scheme that directly predicts the contribution of each expert and generalizes to the broader family of activation functions. We combine the proposed components (sparsity enforcement, expert contribution routing, and dynamic-$k$ gating) into a single method that we call Dense to Dynamic-$k$ Mixture-of-Experts (D2DMoE), which we describe in detail in the next Section.

## 3 Method

D2DMoE reduces the computational cost of the model by splitting every MLP module into a MoE layer. In this section, we describe all of its components in detail. A high-level overview of the entire procedure is presented in Figure 1. The conversion process can be optionally preceded by MHA projection layer replacement (Sec. 3.5), which allows us to apply the same transformation pipeline on all replacement modules.

### 3.1 Enforcing activation sparsity

We expect that enforcing higher levels of activation sparsity may allow for the execution of an even smaller number of experts, resulting in overall computational savings. To this end, we induce activation sparsity by fine-tuning the model with an additional loss term that induces activation sparsity [11]. We apply the *square Hoyer* regularization [22, 17] on the activations of the model:

$$\mathcal{L}_s(x) = \frac{1}{L} \sum_{l=1}^{L} \frac{(\sum_i |a_i^l|)^2}{\sum_i a_i^{l\,2}}, \tag{1}$$

---

[4]We provide the experimental details for this analysis in Section 4.3 and Appendix J.

where $a^l$ is the activation vector from the middle layer of the $l$-th MLP for input $x$, and $L$ is the total number of MLPs in the model. Overall, the model is trained with the following cost function:

$$\mathcal{L}(x) = \mathcal{L}_{\text{CE}}(\hat{y}, y) + \alpha \mathcal{L}_s(x) \tag{2}$$

where $\mathcal{L}_{\text{CE}}$ is cross-entropy loss, and $\alpha$ is the hyperparameter that controls the strength of sparsity enforcement. We find that the pre-trained models recover the original performance with only a fraction of the original training budget (eg. 1B tokens for GPT2-base or Gemma-2B, which is less than 1% of the tokens used for pretraining).

## 3.2    Expert clustering

We split the two-layer MLP modules into experts using the parameter clustering method proposed by Zhang et al. [52]. Assuming the MLP layers are composed of weights $\boldsymbol{W}_1$, $\boldsymbol{W}_2$ and corresponding biases $\boldsymbol{b}_1$, $\boldsymbol{b}_2$, we treat the weights of each neuron from $\boldsymbol{W}_1$ as features and feed them into the balanced $k$-means algorithm [26] that groups neurons with similar weights together. Then, we use the resulting cluster indices to split the first linear layer $\boldsymbol{W}_1$, the first bias vector $\boldsymbol{b}_1$, and the second linear layer $\boldsymbol{W}_2$ into $n$ experts of the same size. The second bias $\boldsymbol{b}_2$ is not affected by this procedure.

MoEfication process was designed for standard two-layered MLPs [52]. Recent LLMs [45, 42] have shifted towards *gated* FFNs, where the activation is realized through a Gated Linear Unit (GLU) [37], which contains an additional weight matrix for the gate projections. To adapt the expert clustering procedure described above to gated FFN layers, we cluster the weights of the gating matrix $\boldsymbol{W_g}$ instead of $\boldsymbol{W_1}$, and use the obtained indices to divide the weights of the two other layers. We provide more intuition and details on our method for gated FFNs in Appendix G.

## 3.3    Expert contribution routing

In a standard MoE-based model, the gating networks are trained in an end-to-end manner. Contrary to this, we train each gating network independently. We propose to frame the problem of training the router as a regression task and directly predict the $\ell^2$-norm of the output of each expert with the router. Formally, given an input token $z$, we train D2DMoE router $R$ to minimize the following loss:

$$\mathcal{L}_r(z) = \frac{1}{n} \sum_i^n \left( R(z)_i - \|E_i(z)\| \right)^2 \tag{3}$$

where $E_i$ is the $i$-th expert. We use a small two-layer neural network as the router $R$ and apply an absolute value activation function to ensure non-negative output. This regression-based formulation is still compatible with the commonly used top-$k$ expert selection, but enables more precise attribution of the contribution of each expert, as we show later in the experimental section.

Note that Zhang et al. [52] also trains each routing network independently, but their method constructs artificial labels for each input, and then subsequently trains the router as a classifier. We discuss the differences in detail in Appendix A.

## 3.4    Dynamic-$k$ gating

Commonly used MoE layers always execute top-$k$ experts for each token, where $k$ is a predefined hyperparameter. This means that, regardless of the difficulty of the input, the model spends the same amount of compute on each batch [54] or token [38]. While this may be appropriate if the model is trained with the same restriction, it is suboptimal for a model that was converted from a dense model, as we show in Section 2.

Since our router directly predicts the $\ell^2$-norm of the output of each expert, we propose a dynamic-$k$ expert selection method that skips experts for whom the router predicts relatively small output norms. Given a router output vector $R(z)$, we select a hyperparameter $\tau \in [0, 1]$ and define the expert selection rule $G$ for the $i$-th element as:

$$G(z)_i = \begin{cases} 1 & \text{if } R(z)_i \geq \tau \cdot \max R(z) \\ 0 & \text{if } R(z)_i < \tau \cdot \max R(z) \end{cases} \tag{4}$$

Note that as $\tau$ increases, the number of executed experts and the overall computational cost decrease. We emphasize that after model deployment $\tau$ can be adjusted without retraining.

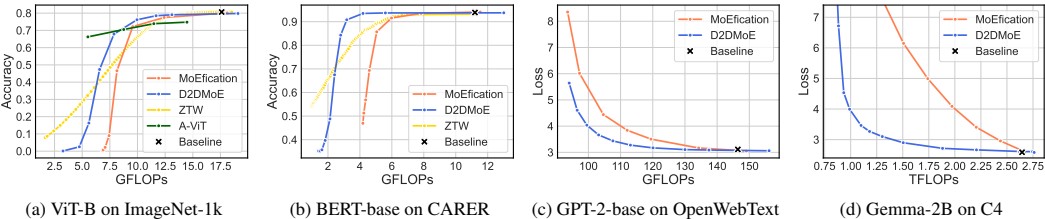

| (a) ViT-B on ImageNet-1k | (b) BERT-base on CARER | (c) GPT-2-base on OpenWebText | (d) Gemma-2B on C4 |

Figure 4: FLOPs-performance tradeoff comparison of our method and MoEfication [52] on CV and NLP benchmarks. We also include early-exit (ZTW, [49]) and token dropping baselines (A-ViT, [51]) for classification. Our method outperforms these baselines across multiple computational budgets.

## 3.5 Conversion of standalone dense layers

A significant amount of computing in deep neural networks is often spent on dense layers that are not followed by any activation function. Dense-to-sparse-MoE conversion methods cannot reduce the costs of such layers due to a lack of activation sparsity. This determines an upper bound on the possible computational savings. To overcome it, we substitute dense layers with small MLPs with approximately the same computational cost and number of parameters. Each MLP is trained to imitate the output of the original dense layer given the same input by minimizing the mean squared error between the two (akin to a distillation loss).

In our case, for Transformer architectures, we substitute projection matrices along with their biases in every MHA layer, as depicted in Figure 3. This means that the final model has four MoE layers in the MHA layer and one MoE layer in the FFN layer (either plain or gated) for each Transformer block. Note that we do not modify the computation of the scaled dot-product attention itself and this scheme can be applied to any standalone dense layer.

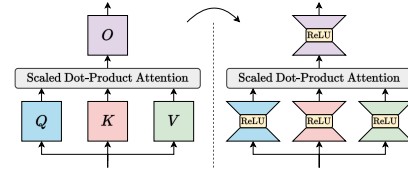

Figure 3: Multi-Head Attention projection conversion scheme.

## 4 Experiments

To analyze the impact of our method, we evaluate its performance on language modeling, text classification, and image classification. We obtain performance versus computational cost characteristics for each method by evaluating the methods with different inference hyperparameters (either $\tau$ described in Section 3.4 for D2DMoE or number of experts $k$ for MoEfication; we mark them on the plots with dot markers). We report the computational cost of each method in FLOPs, as it is a device-independent metric that has been shown to correlate well with latency [27]. In addition, we measure the wall-clock execution time of an efficient implementation of our method.

For MoEfication, we follow the procedure described by Zhang et al. [52] by converting the activation functions of the pre-trained model to ReLU and then fine-tuning the model. In the case of D2DMoE, we also replace activation functions with ReLU, except for Section 5.4, where we demonstrate that our method performs well also with GELU. To provide a fair comparison, the total training data budget is always the same between different methods. See Appendix J for a detailed description of our setup. The source code for our experiments is available at: `https://github.com/bartwojcik/D2DMoE`.

## 4.1 Image classification

Vision Transfomer [7] is one of the most popular architectures in computer vision. Since our method applies to any Transformer model, we evaluate it on the popular ImageNet-1k [35] dataset. We use a pre-trained ViT-B checkpoint as the base model and compare D2DMoE with MoEfication in terms of the computational cost versus accuracy trade-off. For broader comparison, we also evaluate the state-of-the-art early-exit method Zero-time Waste (ZTW) [49], as well as our re-implementation of A-ViT, an efficient token dropping method proposed by Yin et al. [51]. Our results, presented in Figure 4a, demonstrate the significant gains from applying our method over MoEfication.

Table 1: Relative downstream performance of D2DMoE and MoEfication on BoolQ dataset. Our method only starts to degrade at around 70% compute budget, while MoEfication gradually decreases.

| Compute budget | 100% | 90% | 80% | 70% | 60% | 50% | 25% | 10% |
|---|---|---|---|---|---|---|---|---|
| MoEfication | 100% | 92.24% | 92.19% | 92.15% | 88.79% | 75.40% | 86.70% | 77.53% |
| **D2DMoE** | **100%** | **99.68%** | **99.37%** | **98.69%** | **97.60%** | **94.34%** | **92.75%** | **90.89%** |

## 4.2 Text classification

We evaluate our method with BERT-base [6] on the CARER dataset [36] that contains text samples categorized into 6 different emotion categories. We compare the accuracy versus FLOPs trade-off for D2DMoE, MoEfication, and ZTW. We show the results in Figure 4b. The performance of MoEfication gradually deteriorates and completely collapses when the number of executed experts approaches zero. In comparison, D2DMoE maintains the original performance for a wide range of computational budgets, and the performance drop starts at a significantly lower budget. While early-exiting performs well for the lowest budgets, it obtains worse results than D2DMoE at medium budgets and suffers from a gradual performance decline.

## 4.3 Language modeling

We evaluate our method on language modeling and compare it with MoEfication using GPT-2-base [31] and Gemma-2B [42]. We initialize GPT-2 models from a publicly available OpenAI checkpoint pre-trained on a closed-source WebText dataset and use OpenWebText [12] in all of our experiments. For Gemma-2B, we also start from the publicly available pretrained model and evaluate its language capabilities on the C4 dataset [32] after finetuning. For both models, we use around 1B tokens for the finetuning phase (less than 1% of the cost of original pretraining) and 8-16M tokens for router training. We report the results in this section without the MHA projection replacement, as this task is highly sensitive to changes in attention layers, leading to noticeable loss degradation. For more training details, see Appendix J.3

We present test losses for D2DMoE and MoEfication at different compute budgets for GPT-2-base and Gemma-2B in Figures 4c and 4d respectively. Our method outperforms the baseline at every computational budget. The loss of D2DMoE plateaus for higher budget levels, while the baseline displays consistently worse results whenever we lower the computational budget. Notably, for the larger Gemma-2B model our method performs well for most compute budgets, while the performance of MoEfication collapses. The failure of MoEfication can be explained by the emergence of massive activations in large models [40], which makes it unable to learn reliable routing, as we analyze in more detail in Appendix E.

We also provide a downstream evaluation of our Gemma models on the BoolQ dataset. We take the base model, which achieves 68.40% zero-shot evaluation accuracy, and convert it to MoE with D2DMoE and MoEfication. In Table 1, we report the relative accuracy of the models at different compute budgets. Our method largely retains the performance across multiple compute budgets, while the performance of MoEfication decreases significantly. This shows that the loss-vs-FLOPs results for D2DMoE and MoEfication directly translate to downstream performance on language tasks.

## 4.4 Execution latency

For any model acceleration method to be practically useful, it must reduce end-to-end inference execution time on modern GPU hardware. To achieve this, we implement the forward pass of our MoE-layer in the Triton intermediate language [43], and employ several optimizations for our implementation, including an efficient memory access pattern, kernel fusion, and configuration auto-tuning. As suggested by Tan et al. [41], our implementation also avoids unnecessary copies when grouping tokens.

We verify the performance of our implementation for a single D2DMoE layer (24 experts with expert dimensionality 128) layer

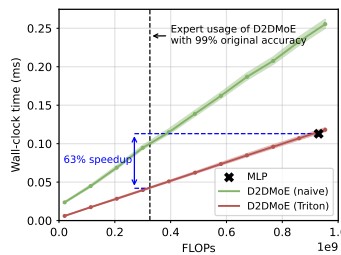

Figure 5: Single D2DMoE layer execution wall-clock time.

in isolation by comparing it with the corresponding MLP module (inner dimensionality 3072) on an NVIDIA A100 GPU. We fill a tensor of size $[256 \times 197 \times 768]$ (batch size, sequence length, and hidden dimension, respectively) filled with Gaussian noise and use it as input to both modules. The gating network of D2DMoE is included in measurements, but the decisions are overridden with samples from a Bernoulli distribution, and we control how many experts are executed on average by changing the Bernoulli probability. The results, presented in Figure 5, show that our implementation scales linearly with the number of executed experts, and has negligible overhead. Our method can be almost three times as fast as standard MLP while preserving 99% of the original accuracy. In Appendix C we provide additional wall-clock measurement results along with a more detailed description of our implementation.

## 4.5 Compatibility with model compression techniques

To accelerate inference D2DMoE leverages input-dependent activation sparsity, a property inherent to almost every Transformer model. However, interaction between D2DMoE and other popular network acceleration techniques, such as pruning [16] or quantization [13, 28], is unclear. We evaluate D2DMoE in combination with such techniques to demonstrate their complementarity.

First, we evaluate D2DMoE applied on top of networks pruned with CoFi, a structured pruning technique introduced by Xia et al. [50]. CoFi removes redundant neurons, attention heads, and sublayers to achieve the desired sparsity ratio, and then subsequently fine-tunes the reduced network. We first prune the base model with CoFi to the desired sparsity level, apply D2DMoE to it, and then evaluate both models on QNLI [48]. In Figure 6, we show that D2DMoE successfully accelerates inference even on networks pruned to high sparsity levels.

In Figure 7, we also investigate the applicability of D2DMoE to quantized models using dynamic post-training quantization from PyTorch[5] on BERT trained on the CARER dataset. Our method is robust to 8- and 16-bit quantization and exhibits only slight variations in performance after quantization. As FLOPs do not take bit width into account, we show quantized models in the same FLOPs range as the original model. In Appendix C, we also present wall-clock time measurements for quantized D2DMoE.

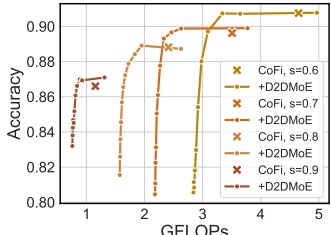

Figure 6: D2DMoE applied to models pruned with CoFi.

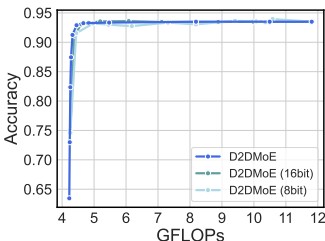

Figure 7: D2DMoE applied to quantized models.

# 5 Analysis

In this section, we present in detail additional experiments that provide insights into the performance of our method. Additionally, in Appendix E we analyze the performance of MoEfication with Gemma, in Appendix F we provide the results of router architecture analysis, in Appendix H we conduct experiments corresponding to the ones in Section 5.5 with GELU function, and in Appendix I we show additional visualizations for expert activation patterns.

## 5.1 Expert selection patterns

The dynamic-$k$ rule introduces variability in the allocation of the computational budget along the model depth. To explore its scale, we investigate the distribution of the number of executed experts, with and without the activation sparsification phase. In Figure 8a, we show the histograms of the number of activated experts for each FFN layer of the BERT-base model trained on the CARER dataset (additional results are available in the appendix in Appendix I). As expected, the model with enforced activation sparsity requires fewer experts for a given threshold. Both base and sparsified models exhibit significant variance in the number of activated neurons across different layers, which justifies the dynamic-$k$ selection and indicates that computational adaptability mechanisms are crucial for efficient inference in Transformer-based models.

---

[5] https://pytorch.org/tutorials/recipes/recipes/dynamic_quantization.html

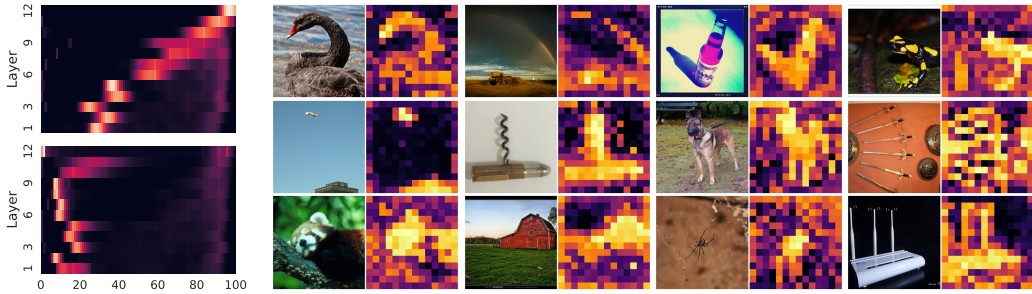

(a) Compute along the model depth

(b) Computational load maps for ImageNet-1k sample images

Figure 8: D2DMoE allows for a dynamical allocation of computation for each layer and each input independently. a) Per-layer distribution of the number of executed experts on CARER dataset in D2DMoE with $\tau = 0.01$ for a standard model (top) and a sparsified model (bottom). Sparsification leads to a significantly lower number of selected experts. b) Computational load maps of selected ImageNet-1k samples for our converted ViT-B model with $\tau = 0.0025$. D2DMoE allocates its computational budget to semantically important regions of the input.

D2DMoE also allows the model to allocate different computational resources to various layers. We expect the model to allocate more compute to tokens containing information relevant to the current task. Since each token position in a ViT model corresponds to a separate and non-overlapping part of the input image, we can easily plot a heatmap to indicate the regions of the image where the model spends its computational budget. In Figure 8b we present such an analysis for our converted ViT-B model. As expected, the dynamic-$k$ routing enables the model to minimize the computational effort spent on regions that contain insignificant information.

## 5.2    Ablation study

Since our method consists of several steps, the positive impact of each one of them may not be evident. To show the significance of every component, we perform an ablation study by incrementally adding each component to the baseline method. We take a BERT-base model and evaluate the ablated variants in the same setting as described in Section 4.2. The results of this experiment are presented in Figure 9a. As expected, each ablated version of the method improves upon the previous one. The sparsity enforcement phase leads to enhanced performance compared to plain MoEfication. Alternative router training objective and dynamic-$k$ expert assignment further improve the results, but – as the method only operates on the FFN layer – the computational cost cannot go below the cost of the remaining part of the model. Extending D2DMoE to MHA projection layers allows our method to reduce the computational cost further, and the resulting full method retains the accuracy of the original model at around twice fewer FLOPs than MoEfication.

## 5.3    Base model activation sparsity

To justify our proposed activation sparsity phase, we investigate the impact of the activation sparsity of the base dense model on the performance MoE obtained with our method. We conduct a study similar to the one presented in Figure 2a: we train multiple base models with different activation sparsity enforcement loss weights $\alpha$ and convert them to Mixture-of-Experts models with our method.

The results, shown in Figure 9b, highlight the positive correlation between the activation sparsity and the performance of the converted MoE, as higher sparsity in the base model always translates to better performance for D2DMoE. This is consistent with results previously observed for MoEfication. However, our method achieves better results for every base model in all cases, proving that regression routing and dynamic-$k$ selection better utilize the induced sparsity.

## 5.4    Sparsification and reliance on the activation function

Activation sparsity works focus their analysis on networks with ReLU activation, as other functions (such as GELU or SiLU) do not guarantee exact sparsity. When analyzing non-ReLU models, such

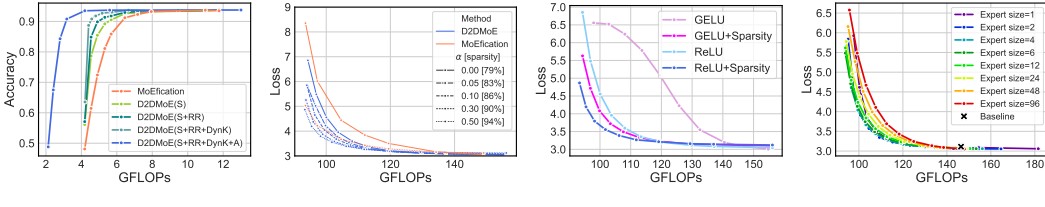

Figure 9: Analysis experiments with D2DMoE. (a) Impact of different phases of our method. Each phase improves upon the baseline. (b) Sparsification improves the cost-accuracy trade-off of the final D2DMoE model. (c) Sparsification allows us to apply our method to GELU-based model without significant drops in performance. (d) Smaller experts display favorable performance and allow for larger computational savings.

works require fine-tuning with the activation function changed to ReLU (*relufication*) [52, 27], which limits their practical applicability. We hypothesize that relufication is not necessary and the models with many near-zero activations effectively function similarly to standard ReLU-based models. To evaluate this hypothesis, we extend the sparsity enforcement scheme to the commonly used GELU activation by penalizing the model for pre-activation values larger than a certain threshold. We first transform the pre-activation values as $z' = \max(0, z - d)$, where $z$ is the pre-activation value and $d$ is a displacement hyperparameter. Then, we apply the loss from Equation (1) on $z'$. This transformation penalizes only pre-activation values larger than $d$, and as a result, the model learns to produce values that effectively become negligible post-activation. We empirically find that $d = -10$ works well with GELU as the output below this value is near zero.

To validate our hypothesis, we follow the methodology from Section 4.3 and we train ReLU- and GELU-based GPT-2 with and without sparsity enforcement loss. We show the results in Figure 9c. D2DMoE with a sparsified GELU-based model performs similarly to a sparsified ReLU-based model, while the performance of the non-sparsified GELU-based variant collapses. Within ReLU-based models, the sparsification still enhances the performance of D2DMoE, but the improvements are less drastic, and the behavior of our method does not significantly change as in the case of GELU. This shows sufficient activation sparsity enforcement relieves the model from the dependence on ReLU.

## 5.5 Impact of expert granularity

A crucial hyperparameter in D2DMoE is the selection of expert size. Smaller experts may allow a more granular selection of executed neurons, likely resulting in a lower computational cost. However, decreasing the expert size increases the number of experts, which translates to a larger router, potentially negating any computational gains. To study the impact of this hyperparameter on our method, we evaluate D2DMoE on GPT-2 with different expert sizes, and show the results in Figure 9d.

We observe that our method generally performs better with smaller experts. Those results differ from the ones presented in [52], where the expert size is significantly higher. The positive correlation between granularity and performance can be explained by the increased levels of activation sparsity in our model, which requires significantly fewer activated neurons (experts). As expected, the performance decreases for the extreme choice of expert size equal to 1 due to significantly higher routing costs. We include additional results for expert granularity with GELU activation in Appendix H.

## 6 Related Work

**Mixture-of-Experts.** MoE layers were introduced as an efficient way to further increase the capacity of deep neural networks applied for NLP tasks, initially in LSTM models [38], and later in Transformers [23]. Since then, they have also been applied to computer vision [33, 5]. MoE layers have gained significant popularity primarily due to their excellent scaling properties [8, 3]. Nonetheless, training such models is challenging, primarily because gating decisions must be discrete to ensure sparse expert selection. Various methods of training were proposed, some of which include reinforcement learning [1], weighting the expert output by the probability to allow computation of the gradient of the router [38], or using the Sinkhorn algorithm [3]. Some of those approaches also suffer

from the possibility of load imbalance and therefore require auxiliary losses or alternative expert selection methods [9, 54]. Interestingly, in many cases fixed routing functions perform similarly to trainable routers [34], which suggests that current solutions are largely suboptimal. MoE models can also be derived from pre-trained dense models by splitting the model weights into experts and independently training the routers for each layer [52, 57], which avoids most of the problems present in end-to-end training.

**Activation sparsity in Transformers.** Li et al. [24] show that ReLU-based Transformer models produce sparse activations in their intermediate representations, an effect that is prevalent across architectures, layers, and modalities. They propose a simple rule for keeping only top-$k$ activations in each MLP layer, which results in a model with comparable performance. Similarly, Mirzadeh et al. [27] demonstrate that ReLU activation function in LLMs encourages ensuing activation sparsity that can be leveraged to skip redundant computations. Tuli and Jha [46] take a step further and design a dedicated Transformer architecture accelerator that also exploits activation sparsity, while Liu et al. [25] proposes to predict activation sparsity structure in LLMs and reduce the model latency by skipping redundant computations. Jaszczur et al. [18] demonstrate that it is possible to train Transformer models from scratch with a fixed level of activation sparsity and obtain similar performance. Finally, a related line of works focuses on sparsity in the attention distributions instead of intermediate representations [4]. None of the above-mentioned methods explore induced activation sparsity as a way to increase computational gains, nor do they address variance of the number of sparse activations on a per-token basis.

# 7    Conclusion

We introduce Dense to Dynamic-$k$ Mixture-of-Experts (D2DMoE), a novel approach that induces activation sparsity to improve the efficiency of Transformer-based models by converting their layers to Mixture-of-Experts (MoE). We demonstrate the interplay between the activation sparsity of dense models and the efficiency of converted MoEs. Moreover, we introduce regression-based router training and dynamic-$k$ routing, which enable our method to efficiently utilize the induced sparsity. Finally, we show how dense-to-sparse-MoE conversion approaches can be extended to MHA projections and gated MLPs. Our approach is compatible with the existing Transformer architectures and significantly improves upon existing MoE conversion schemes. Our findings contribute to the ongoing efforts to make Transformer models more efficient and accessible for a wider range of applications, especially in resource-constrained environments.

## Limitations and Broader Impact

While D2DMoE displays promising results in reducing the computational cost of inference in Transformer models, a few limitations should be acknowledged. Our proposed sparsity enforcement and router training phases require additional training time. This overhead, while small, must be considered when evaluating the benefits of our approach. Moreover, we demonstrate improved performance over existing approaches on common NLP and CV tasks, but the scope of our experiments is restricted due to limited access to computational resources. Further research is needed to explore its applicability to extremely large models.

Our work focuses primarily on fundamental machine learning research and we do not see any specific risks or ethical issues associated with our method. Nevertheless, we recognize the potential for misuse of machine learning technology and advocate for responsible AI practices to mitigate such risks.

## Acknowledgments

Filip Szatkowski is supported by National Centre of Science (NCP, Poland) Grant No. 2022/45/B/ST6/02817. Bartosz Wójcik is supported by National Centre of Science (NCP, Poland) Grant No. 2023/49/N/ST6/02513. Simone Scardapane was partly funded by Sapienza grant RG123188B3EF6A80 (CENTS). This paper has been supported by the Horizon Europe Programme (HORIZON-CL4-2022-HUMAN-02) under the project "ELIAS: European Lighthouse of AI for Sustainability", GA no. 101120237. For the purpose of Open Access, the authors have applied a

We gratefully acknowledge Poland's high-performance Infrastructure PLGrid (HPC Centers: ACK Cyfronet AGH, PCSS, CI TASK, WCSS) for providing computer facilities and support within computational grants no. PLG/2023/016393, PLG/2023/016321, and PLG/2024/017385.

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

## A    Difference between router training in D2DMoE and MoEfication

Our router training procedure is similar to the one proposed in MoEfication [52], but the source code of the method provided by the authors[6] contains a different routing objective than the one reported in the paper. While the paper describes their router training objective as a prediction of the sum of ReLU activation values in each expert, the source code uses prediction labels created from the sum of the activations in the intermediate layer divided by the maximum value in the entire batch and minimize the binary cross-entropy loss. Assuming that $a_{k,j}$ is the activation vector in the hidden layer of expert $j$ for sample $k$, the label generation for their router can be expressed as:

$$y_{k,j} = \frac{\sum_i a_{k,j,i}}{\max_{l,m} \sum_i a_{l,m,i}} \tag{5}$$

In comparison to their approach, the router training procedure in D2DMoE differs in multiple aspects:

- Our router considers the output of each expert instead of looking at the activations in the intermediate layers.
- Instead of using artificially created labels based on the sums of activation values, we predict the $\ell^2$-norm of the output. This has the additional benefit that our router can work with alternative activation functions.
- Our router is trained with the mean-squared error instead of the binary cross-entropy loss. The output of our router is constrained to positive values, while the MoEfication router is constrained to outputs in $[0, 1]$.

We find that the above differences are responsible for the improved performance of our router (see Figure 9a).

## B    Comparison of FLOPs between standard FFN layer and dynamic-$k$ MoE

To compare the efficiency of our method with a standard MLP layer in Transformer, we estimate FLOPs in both modules. We assume the layer is composed of two linear transformations, with input and output size $d_m$ and hidden dimension $ed_m$, where $e$ is the expansion factor, which is usually equal to 4 in standard Transformer models. We skip the negligible cost of the biases and activation functions for simplicity.

One can estimate the cost of the MLP layer in FLOPs, $C_{\text{MLP}}$, as:

$$C_{\text{FFN}} = d_m \cdot ed_m + ed_m \cdot d_m = d_m{}^2 \cdot 2e. \tag{6}$$

For dynamic-$k$ expert selection with $n$ total experts and $k$ experts selected for a given input, the cost of the forward pass is composed of the cost of a forward pass through $k$ experts and the cost of the 2-layer router with input dimension $d_m$, hidden dimension $d_h$ and output dimension $n$. The cost of the single expert pass can be expressed as:

$$C_E = \left(d_m \cdot \frac{ed_m}{n} + \frac{ed_m}{n} \cdot d_m\right) = d_m{}^2 \cdot \frac{2e}{n}, \tag{7}$$

and the routing cost can be estimated as:

$$C_R = d_m \cdot d_h + d_h \cdot n. \tag{8}$$

Therefore, the full cost of dynamic-$k$ $C_{\text{dynk}}$ can be estimated as:

$$C_{\text{dynk}} = k \cdot C_E + C_R = d_m{}^2 \cdot \frac{2ek}{n} + d_h(d_m + n), \tag{9}$$

and the cost of our method compared to the cost of standard MLP can be expressed as:

$$\frac{C_{\text{dynk}}}{C_{\text{MLP}}} = \frac{d_m{}^2 \cdot \frac{2ek}{n} + d_h(d_m + n)}{d_m{}^2 \cdot 2e} \tag{10}$$

$$= \frac{k}{n} + \frac{d_h(1 + \frac{n}{d_m})}{d_m \cdot 2e}. \tag{11}$$

---

[6]MoEfication source code for router training is publicly available at: `https://github.com/thunlp/MoEfication/blob/c50bb850307a36f8a0add6123f56ba309a156d13/moefication/utils.py#L188-L260`

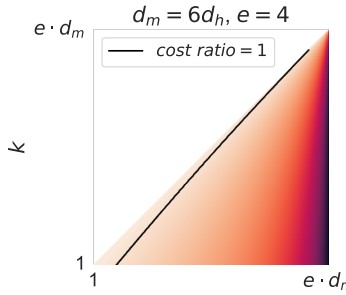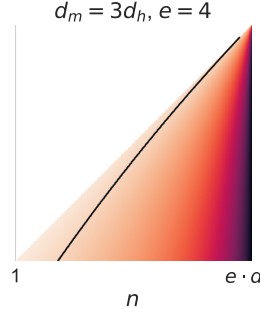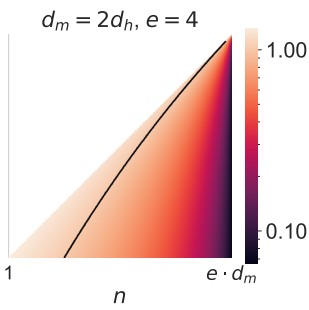

Figure 10: FLOPs ratio between dynamic-$k$ expert layer and standard two-layer MLP for different values of the total number of experts $n$ and number of selected experts $k$. We assume the hidden dimension of router $d_h$ is based on model dimension $d_m$, and set standard expansion factor $e = 4$. For different sizes of router, dynamic-$k$ uses fewer FLOPs than standard MLP as long as the total number of experts is sufficiently large and the number of selected experts is not equal to the total number of experts. For the clarity of presentation, we plot discrete values of $k$ and $n$ as continuous.

As long as the number of selected experts $k$ does not approach the total number of experts $n$ and the hidden dimension of the router does not approach the size of hidden dimension $d_m$, the ratio is significantly below one.

Assuming the worst case for second term ($n = ed_m$), we can estimate the cost ratio as:

$$\frac{k}{n} + \frac{d_h}{d_m} \cdot \frac{1+e}{2e}, \tag{12}$$

which shows that dynamic-$k$ expert selection only exceeds the FLOPs cost of the standard network when the dynamic-$k$ rule selects almost all experts or the number of experts becomes very high. For an even more detailed analysis, we refer to Figure 10 where the cost ratio between our method and standard MLP is shown, assuming different router sizes and $e = 4$ as standard for most Transformer models. In practice, we use $d_h = 128$, so in all our experiments $d_m = 6d_h$.

## C Efficient implementation of D2DMoE

In Listing 1 we present the pseudocode for our efficient implementation of the forward pass of the D2DMoE module. We skip the pseudocode of the kernel of the second layer as it is similar, but provide the full source code in our code repository. Note that our implementation has multiple points where it could be improved for further performance gains: 1) metadata that is required for the kernels could also be computed with a dedicated kernel to reduce overhead; 2) atomic operations are currently used in the second layer to merge the results from different experts, an alternative implementation that does not use atomic operations could be faster; 3) it could be rewritten in CUDA to make use of dynamic parallelism. We leave those improvements for future work.

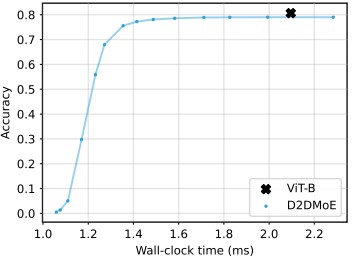

Figure 11: Wall-clock time measurements of the ViT-B model and its corresponding D2DMoE model.

In the main paper, we have presented wall-clock time measurements of a single D2DMoE layer. Below, we also ensure that our implementation works and performs well when used for the ViT-B model in which each FFN is replaced with a D2DMoE module. In Figure 11, we measure the averaged processing time and the accuracy of our model. We perform the experiments on an NVIDIA A100 GPU using a batch size of 256. Each point on the x-axis corresponds to a single $\tau$ threshold and shows the wall-clock time of processing a single input averaged over the entire ImageNet-1k test set. Dynamic inference with D2DMoE offers up to 30% reduction in processing time without affecting the accuracy.

To show that D2DMoE also reduces the execution latency of quantized models, we modify our kernels to handle `float16` and `int8` data types. In Table 2 we perform a similar experiment to the

one from Figure 5. We sample gating decisions from the Bernoulli distribution with probability $p$ and measure the execution time of our experts for the three data type variants.

Table 2: Wall-clock time measurements ($\mu$s) of execution of our D2DMoE layer when using different data types and GPUs.

| **GPU** | $p$ | **0.0** | **0.1** | **0.2** | **0.3** | **0.4** | **0.5** | **0.6** | **0.7** | **0.8** | **0.9** | **1.0** |
|---|---|---|---|---|---|---|---|---|---|---|---|---|
| | `float32` | 5 | 9 | 13 | 18 | 23 | 28 | 33 | 38 | 42 | 47 | 52 |
| RTX 4090 | `float16` | 4 | 5 | 7 | 9 | 11 | 14 | 16 | 18 | 21 | 24 | 27 |
| | `int8` | 4 | 4 | 5 | 7 | 8 | 9 | 10 | 11 | 12 | 13 | 14 |
| | `float32` | 6 | 9 | 12 | 15 | 19 | 22 | 25 | 28 | 31 | 35 | 38 |
| A100 | `float16` | 6 | 7 | 8 | 10 | 11 | 13 | 14 | 16 | 17 | 19 | 21 |
| | `int8` | 7 | 8 | 9 | 10 | 11 | 12 | 14 | 15 | 16 | 17 | 19 |

The results show that both the higher activation sparsity (lower $p$) of our method and lower-precision data types are complementary in terms of wall-clock time reduction. While we see a smaller improvement from using `int8` over `float16` on A100, we attribute this to differences between GPU architectures and software support for low-precision arithmetic.

## D   Compatibility with knowledge distillation

In Section 4.5 we have demonstrated that our method is compatible with two popular model compression methods: quantization and pruning. A natural question is whether our method can be effectively applied to models compressed via knowledge distillation. Since distilled models also exhibit activation sparsity that our method relies on, D2DMoE should be applicable to such models. In Figure 12 we demonstrate the results of D2DMoE when applied on a ViT-S model, which has been trained via knowledge distillation [15] with the torchvision ViT-B being used as the teacher model. We see that D2DMoE is also able to reduce the cost of this smaller model.

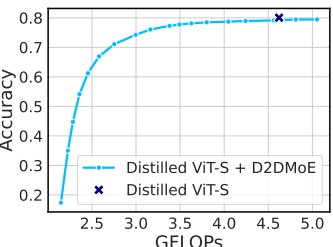

Figure 12: Performance of D2DMoE applied on a ViT-S distilled from the larger ViT-B model.

## E   Routing analysis for large models

As presented in Figure 4d, in comparison to other considered benchmarks MoEfication visibly underperforms on language modeling with Gemma-2B. We attribute this to the emergence of massive activations in LLMs that reach a specific scale [40]. Massive activations are outliers along certain feature dimensions whose magnitudes are thousands of times larger than the magnitudes of other activations. The training objective of MoEfication described in Equation (5) uses maximum activation over the entire batch to normalize the target label for each expert. Upon encountering large outlier values, those labels become effectively meaningless, as the values for most of the experts become very close to zero. In this case, the router effectively learns to output zero labels for most of the experts aside from the ones corresponding to the outlier values.

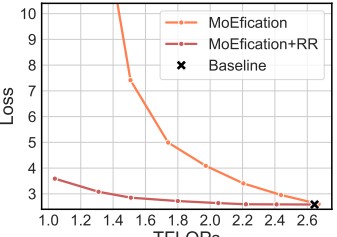

Figure 13: Comparison of performance on Gemma-2B for MoEfication with vanilla routing and with our regression routing.

In comparison to MoEfication, our router training scheme does not make use of such normalization, and should therefore be robust to the emergence of massive activations. To validate this, we apply MoEfication on Gemma-2B, but with our regression routing instead of the original router training strategy. We compare the resulting model with vanilla MoEfication in Figure 13 and notice that replacing the routing scheme is enough for the model to learn effective expert assignment, as even though the expert choice is static and the base model is not sparsified, the cost-loss trade-off has

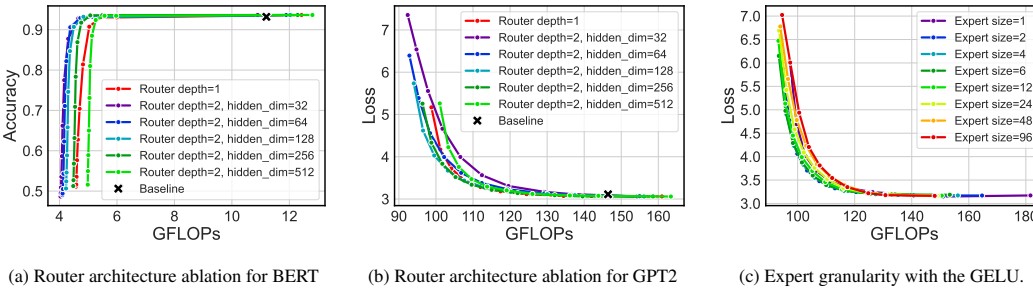

(a) Router architecture ablation for BERT     (b) Router architecture ablation for GPT2     (c) Expert granularity with the GELU.

Figure 14: Additional ablations with router architecture and expert granularity.

significantly improved. This simple experiment shows that our regression routing objective is more robust than MoEfication when scaling to larger models.

## F    Router architecture

In comparison to standard linear routers used in MoE models trained from scratch, routers in MoEfication are 2-layer MLPs. To obtain the best performance with D2DMoE, we compare the linear design with MLPs with different hidden sizes for BERT-base and GPT-2-base on Figures 14a and 14b respectively. Linear routers do not perform well with our method, and overall a 2-layer MLP with a hidden dimension of 128 results in the best performance for both models. Note how for BERT-base, the accuracy curve for a model with the hidden dimension of 128 is slightly worse than for smaller routers, but for harder task with GPT-2 a more complex router is required. Following this analysis, we use 2-layer MLP with a hidden dimension of 128 for most of our experiments in the paper, with the only exception being the larger Gemma-2B model where we scale the hidden dimension accordingly to 512 to match the increase in model dimensionality.

## G    D2DMoE extension to GLU-based layers

To provide better intuition behind the extension of our method to GLU-based gated MLPs mentioned in Section 3.2, we visualize the differences between standard FFN and Gated FFN and the application of our method in Figure 15. Standard Transformer MLP realizes the following function:

$$y(x) = \mathbf{W}_1 A(\mathbf{W}_2 x), \qquad (13)$$

where $\mathbf{W}_1$, $\mathbf{W}_2$ are the weights for the upscale and downscale projections[7] and $A$ stands for the activation function. In comparison, gated MLP can be written down as:

$$y(x) = \mathbf{W}_1(A(\mathbf{W}_g x) \circ \mathbf{W}_2 x), \qquad (14)$$

where $\mathbf{W}_g$ is the weight for the added gate projection.

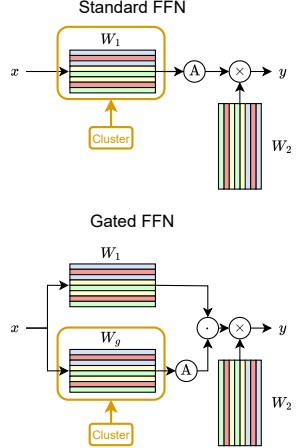

Figure 15: D2DMoE extension to Gated MLP.

The intuition behind MoEfication, which our method also follows for standard FFNs, is that the sparsity of the intermediate, post-activation representations determines the sparsity of the output representation. Therefore, the expert split is performed based on the weights of the upscale projection, as zeroed neurons in the upscale activations will also result in zeroed outputs of the downscale projection. When extending D2DMoE to Gated MLPs, our intuition is that the gating projections determine the sparsity of all the later representations, as both upscale and downscale are multiplied with the gating values. Therefore, we propose to build the experts through clustering performed on the gating weights $\mathbf{W}_g$ and use the indices obtained through expert split on gating weights to construct experts from $\mathbf{W}_1$ and $\mathbf{W}_2$. Following similar reasoning, for GLU-based models, we also perform activation sparsity enforcement on the gating projections instead of upscale projections as described originally in Section 3.1.

---

[7]We omit biases for simplicity.

# H  Additional results with expert size and GELU

In addition to experiments in Section 5.5, we present the results of similar ablation carried on the sparsified GPT-2 model with GELU activation. The results, presented in Figure 14c, follow the same pattern as before, which supports our claim that the sparsification enables the GELU-based models to function similarly to ReLU-based ones.

# I  Expert activation patterns for attention projection layers

Following the analysis for MoE-converted FFN layers in Section 5.1, we present full results for FFN in Figure 16, and investigate the activation patterns in MHA projections modified with our method in Figures 17 to 20. The projection modules display lower levels of sparsity than FFNs, which is to be expected as our projection layers have lower intermediate dimensionality. Expert selection distribution patterns in $Q$ and $K$ show significant similarity, and the patterns in $V$ and output projections are also similar to a lesser degree. The variance of the number of selected experts in MHA projections is higher than in FFN layers, but it still exists and the distribution in some of the layers seems to be bimodal, which provides further justification for the dynamic-$k$ selection rule.

# J  Training and hardware details

In this Section, we describe the technical details used in the D2DMoE conversion procedure. For full reproducibility, we share the source code that we used for conducting the experiments. All experiments were performed using the `PyTorch` library [29] on the NVIDIA A100 and V100 GPUs on internal clusters. We utilize the *fvcore* library to count model FLOPs[8].

## J.1  Image classification

All methods start with the same pre-trained ViT-B from the `torchvision` [9] library and are trained on ImageNet-1k using the augmentation proposed by Touvron et al. [44]. We use mixup $(0.8)$, cutmix, label smoothing $(0.1)$, gradient clipping $(1.0)$ and the Adam optimizer with a cosine learning rate schedule without warm-up. For D2DMoE, we replace the MHA projections and train the replacements for 3 epochs with the initial learning rate $0.001$ and batch size $128$, and then finetune the model for 90 epochs with sparsity enforcement weight $\alpha = 0.2$, initial learning rate $2 \cdot 10^{-5}$ and batch size $512$. We then convert the modules into MoE layers, and train the gating networks for 7 epochs with the initial learning rate set to $0.001$ and batch size $128$. We train ZTW for 100 epochs in total, allocating 5 epochs for ensemble training, while keeping the rest of the original hyperparameters unchanged. For MoEfication, we first convert the pre-trained model to ReLU-based one and finetune for 90 epochs with an initial learning rate of $0.0001$ and batch size $256$. We then split the weights and train the routers for 10 epochs with the initial learning rate $0.001$ and batch size $256$.

## J.2  Text classification

All experiments start from the same pre-trained BERT-base checkpoint. For methods requiring ReLU activation function, we replaced GELU with ReLU and continue model pretraining on concatenated wikipedia [10] and books [55] corpora for $5000$ steps on $8$ GPUs using main setup from `https://github.com/huggingface/transformers/blob/main/examples/pytorch/language-modeling/run_mlm.py`, per device batch size $96$ and learning rate $5 \cdot 10^{-4}$. For MHA projections replacement we use the same corpus and train replaced MLP modules on a single GPU with batch size $128$ and learning rate $0.001$ for $3000$ steps. We finetuned base dense models on CARER dataset for $5$ epochs with $2 \cdot 10^{-5}$ learning rate. For sparsity enforcement in D2DMoE we use $\alpha$ linearly increasing from zero to $0.0001$ over training. For both MoEfication and D2DMoE we train routers with batch size $64$ and initial learning rate $0.001$ for $5$ epochs. In all experiments, we use Adam optimizer with linear learning rate decay. For MoEfication we use expert size $32$, for D2DMoE we use more granular expert size equals $6$. For ZTW we trained ICs for $5$ epochs with batch size $32$ and learning rate $0.01$.

---

[8]`https://github.com/facebookresearch/fvcore`
[9]`https://pytorch.org/vision/stable/models.html`

### J.3 Language modeling

We base our code and hyperparameters for GPT2-base on the *nanoGPT* repository provided at `https://github.com/karpathy/nanoGPT`. We initialize the model from `https://huggingface.co/openai-community/gpt2`. In all pretraining experiments, we initialize models from a publicly available OpenAI checkpoint pre-trained on a closed-source WebText dataset and finetune for the fixed number of 1000 steps with the effective batch size equal to the value in the repository through gradient accumulation. The alpha values for sparsity enforcement can be found at Figure 9b. We train the routers for D2DMoE and MoEfication for 2000 steps using one GPU and tuning the learning rates for a given expert size from the range between $0.002 - 0.005$. For router training, we use Adam optimizer and cosine warmup scheduler.

For Gemma-2B, we start from the checkpoint at `https://huggingface.co/google/gemma-2b`. We also finetune the model for 1k steps with an effective batch size of 1024, sequence length of 1024 and Adam optimizer with a learning rate of 1e-4. As Gemma's hidden dimension is much larger than the other considered models, we change the hidden dimensionality of the routers to 512 for both our method and MoEfication, but keep the other hyperparameters the same as in the rest of the experiments. For MoEfication, Gemma, we use 512 experts to obtain an expert size comparable to the one in their paper. For our method, we use 2048 experts. In D2DMoE, we set sparsity enforcement weight to $0.00003$. We train the routers for 500 steps with Adam and effective batch size of 16 and use a learning rate of $0.001$.

We report the results for language modeling without the MHA projection replacement step, as we find that it is especially sensitive to changes in the attention layers, which always result in visible loss degradation.

## K Contributions

Filip integrated the codebase and ran the experiments for GPT-2 and Gemma, performed the activation sparsity analysis, and all the analyses on language modeling models. He contributed to the design of dynamic-k gating and played a primary role in designing the experiments and writing the article.

Bartosz set the research direction of the project and proposed the alternative routing scheme, dynamic-k expert selection, the additional activation sparsity enforcement phase for ReLU and GELU, and the replacement of MHA projection layers. He wrote the shared codebase for the experiments, carried out the ViT-B experiments, implemented the custom Triton kernels for the efficient implementation of the method, and also played a primary role in the writing and editing of the article.

Mikołaj made this paper possible by performing all of the experiments at the initial stages of the project and implementing MoEfication and numerous variants of our method. He carried out the BERT experiments, performed weight sparsity compatibility analysis, the ablation study, and contributed to the crafting of the paper.

Simone significantly improved the paper's readability and provided invaluable advice for revising it.

```python
def forward_triton_atomic(self, x, routing_tensor):
    # compute the necessary metadata
    # split the batch into two groups: executed by that expert or not
    # (for each expert independently)
    sort_indices = routing_tensor.argsort(dim=0, descending=True)
    # get the number of samples executed by each expert
    expert_bincounts = routing_tensor.sum(dim=0)
    # actual forward pass
    intermediate_acts = MoeFirstLayerImplementation.apply(...)
    final_out = MoeSecondLayerAtomicImplementation.apply(...)
    return final_out

class MoeFirstLayerImplementation(torch.autograd.Function):
    @staticmethod
    def forward(input, weight, bias, sort_indices, expert_bincounts):
        ...
        # a grid of kernel instances which divide the computational work
        # in multiple dimensions: batch dimension (sample_dim),
        # output dimension (expert_dim) and number of experts dimension (num_experts)
        grid = (cdiv(sample_dim, BLOCK_SIZE_BD) *
                cdiv(expert_dim, BLOCK_SIZE_ED), num_experts)
        moe_first_kernel[grid](...)
        ...

@triton.jit
def moe_first_kernel(x_ptr, ...
                     weight_ptr, ...
                     bias_ptr, ...
                     output_ptr, ...
                     sort_indices_ptr, ...
                     expert_bincounts_ptr,
                     ...,
                     ):
    # based on tl.program_id(axis=0), compute the tile indices
    # for the batch and output dimensions
    # (grouped, column major or row major ordering)
    pid_bd, pid_ed = ...
    # kernel instances and experts have a many-to-one relationship
    expert_index = tl.program_id(axis=1)
    # load the total number of tokens assigned to this expert
    expert_samples_count = tl.load(expert_bincounts_ptr + expert_index)
    # calculate the number of instances that need to be used to process all tokens
    bd_pids_for_expert = tl.cdiv(expert_samples_count, BLOCK_SIZE_BD)
    # instances that have no computation to perform exit early
    if pid_bd < bd_pids_for_expert:
        # calculate offsets that will be used for addressing data in memory
        offs_bd = ...
        offs_ed = ...
        offs_hd = ...
        # pick the data to load based on the sort indices
        in_data_indices = tl.load(sort_indices_ptr + expert_index * ... + offs_bd * ...)
        # calculate memory addresses of the input data and weights
        # during loading this will group samples for the current learner only
        x_ptrs = x_ptr + in_data_indices[:, None] * ...
        w_ptrs = weight_ptr + expert_index * ...
        # the result will be accumulated in this variable
        accumulator = tl.zeros((BLOCK_SIZE_BD, BLOCK_SIZE_ED), dtype=tl.float32)
        # iterate over the innermost dimension
        for k in range(0, tl.cdiv(hidden_dim, BLOCK_SIZE_HD)):
            # load the memory from the current input and weight tiles
            x = tl.load(x_ptrs, mask=..., other=0.0)
            w = tl.load(w_ptrs, mask=..., other=0.0)
            # perform matrix multiplication for these tiles and accumulate
            # (since data is grouped, this can be performed in an efficient manner)
            accumulator += tl.dot(x, w)
            # advance the pointers to the next tile
            x_ptrs += BLOCK_SIZE_HD * stride_x_hd
            w_ptrs += BLOCK_SIZE_HD * stride_weight_hd
        # load and add biases to the accumulated result
        offs_b_ed = ...
        b_ptrs = bias_ptr + expert_index * ...
        accumulator += tl.load(b_ptrs, mask=..., other=0.0)
        # apply the activation function on the result
        if ACTIVATION == 'relu':
            accumulator = relu(accumulator)
        ...
        # calculate the memory addresses for the output
        offs_out_bd = ...
        out_ptrs = output_ptr + expert_index * ... + \
                   offs_out_bd[:, None] * ... + offs_b_ed[None, :] * ...
        out_mask = ...
        # store the result to the main GPU memory
        tl.store(out_ptrs, accumulator, mask=out_mask)
```

Listing 1: Simplified pseudocode of our efficient D2DMoE implementation for GPUs

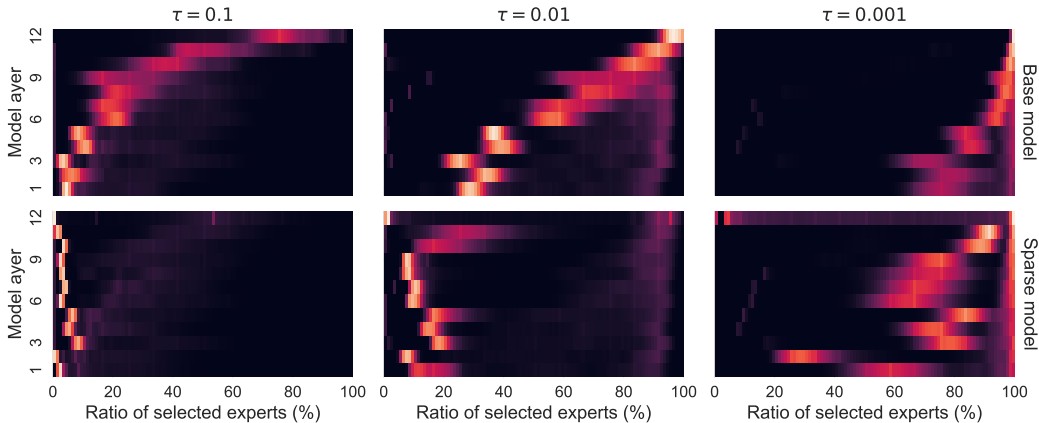

Figure 16: Per-layer distribution of the number of executed experts in D2DMoE trained on the CARER with different $\tau$ thresholds for a standard, non-sparsified model (top row) and a sparsified model (bottom row). The high variability of that number explains the computational gains from using dynamic-$k$.

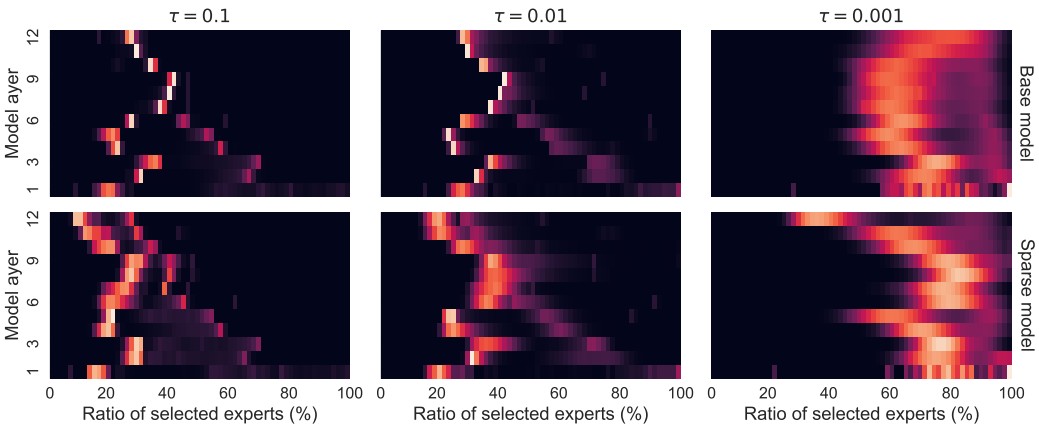

Figure 17: Distribution of the number of executed experts in each layer for query projections.

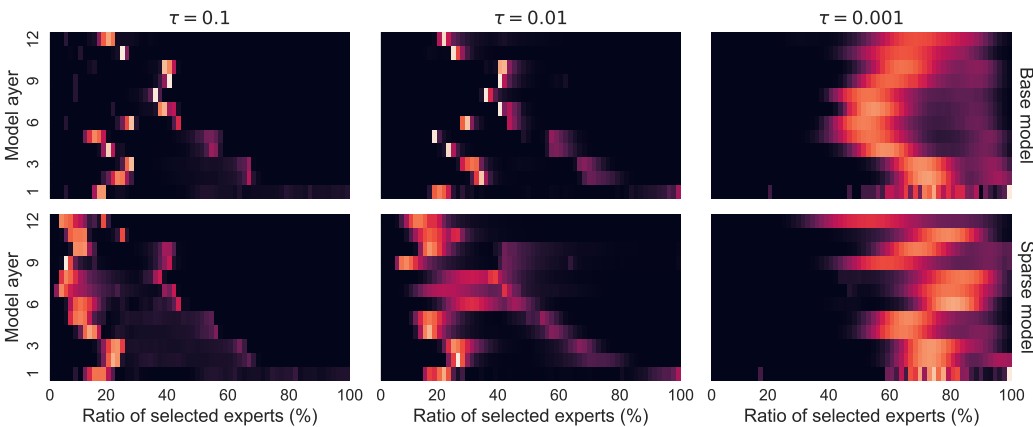

Figure 18: Distribution of the number of executed experts in each layer for key projections.

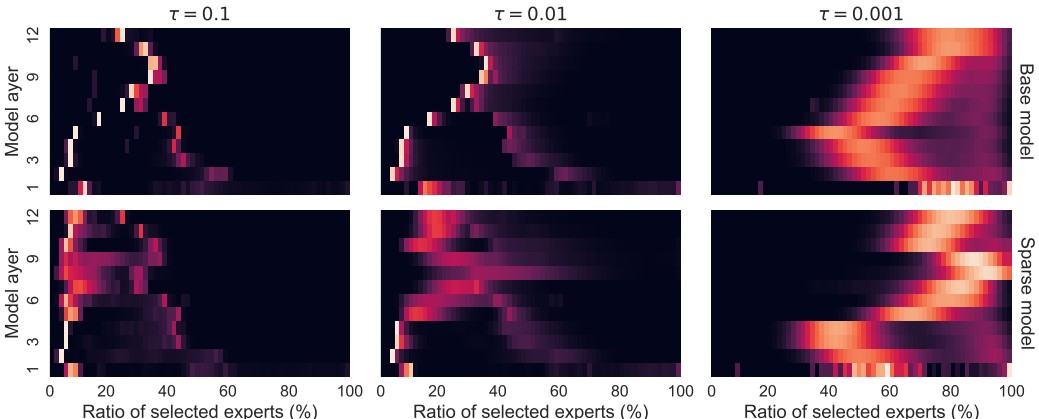

Figure 19: Distribution of the number of executed experts in each layer for value projections.

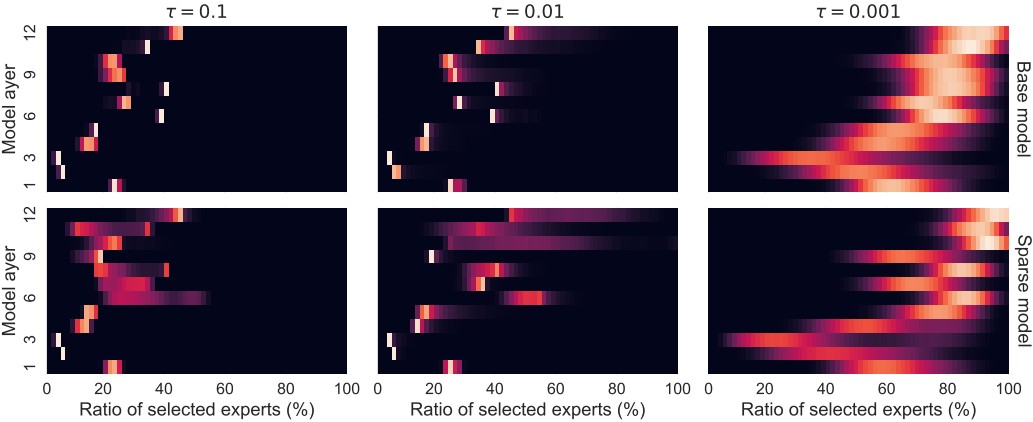

Figure 20: Distribution of the number of executed experts in each layer for output projections.

