# OpenReview forum: "Exploiting Activation Sparsity with Dense to Dynamic-k Mixture-of-Experts Conversion"
_NeurIPS.cc/2024/Conference — NeurIPS 2024 poster_

### Official Review · Reviewer_85BP · 2024-07-13

**Soundness:** 2
**Presentation:** 3
**Contribution:** 2
**Rating:** 3
**Confidence:** 4

**Summary:**

This paper proposes a more effective dynamic-k expert selection rule that adjusts the number of executed experts on a per-token basis to reduce the high computational cost of the transformer models. They claim their D2DMoE model outperforms existing approaches.

**Strengths:**

1. The motivation of this paper is good. They try to save up the inference cost of the transformer models.
2. They perform experiments on several tasks.

**Weaknesses:**

1. The most important contribution is the dynamic-k routing algorithms. However, the main work is a small modification to MoEfication. The difference of the proposed method with MoE-based methods is limited.
2. The experiments are not conducted well. This paper lacks a comprehensive comparison with other methods. There exist many MoE-based VIT methods, such as [1, 2]. Furthermore, why does the authors does not provide any performance number in the table? This paper claims they outperforms existing approaches on common NLP and vision tasks, however, only two methods are compared.
3. The major advantage of D2DMoE compared with MoEfication is saving inference cost. However, the saved cost is quite limited observed from fig. 4.

[1] Mod-squad: Designing mixtures of experts as modular multi-task learners
[2] M3 vit: Mixture-of-experts vision transformer for efficient multi-task learning with model-accelerator co-design

**Questions:**

1. Training Overhead: Can you provide more details on the additional training time required for sparsity enforcement and router training? How does this overhead compare to the overall training time of the original dense models?
2. Real-world Deployment: What steps or considerations are necessary for deploying D2DMoE models in real-world applications, particularly in terms of latency, hardware compatibility, and integration with existing pipelines?
3. Adversarial Robustness: Have you evaluated the robustness of D2DMoE models against adversarial attacks? If not, do you have any plans to explore this aspect in future work?
4. Comparison with Pruning Techniques: How does D2DMoE compare with other model pruning or compression techniques in terms of performance, computational savings, and implementation complexity?
5. Scalability: Can the proposed method be effectively scaled to extremely large models like GPT-3? Are there any specific challenges or considerations when applying D2DMoE to such models?

**Limitations:**

The paper does not sufficiently address potential challenges in real-world deployment, such as latency variations, hardware compatibility, and integration with existing systems.

---

> ### Author Rebuttal · Authors · 2024-08-06
>
> We thank the reviewer for the time spent on our work. We are glad that the reviewer recognizes the proper motivation of our method and the thoroughness of our experiments. Below, we address the concerns raised by the reviewer.
>
> ### Weaknesses
>
> > The most important contribution is the dynamic-k routing algorithms. However, the main work is a small modification to MoEfication. The difference of the proposed method with MoE-based methods is limited.
>
> The contributions of our work are not only dynamic-k gating, but also the sparsification phase, expert contribution routing, and extension to MHA projections. All these contributions improve upon the baseline method (Figure 6a). Figures 2a and 6b show the impact of the sparsification phase on the model performance. Regression routing significantly improves the performance on Gemma2B, even when applied in isolation to MoEfication (see Appendix C), while the baseline method performs very badly in this setting. This indicates our method is more robust when scaling to larger models (note how our routing significantly differs from the MoEfication objective, as explained in Appendix A).
>
> > The major advantage of D2DMoE compared with MoEfication is saving inference cost. However, the saved cost is quite limited observed from fig. 4.
>
> We respectfully disagree with the reviewer’s claim that the cost savings with our method are insignificant. Note how our method matches the performance of the dense model using around 60% (ViT), 30% (BERT), 80% (GPT-2) or 70% (Gemma2B) of its computation, significantly reducing the cost when compared to the MoEfication baseline.
>
> > The experiments are not conducted well. This paper lacks a comprehensive comparison with other methods. There exist many MoE-based VIT methods, such as [1, 2].
>
> > This paper claims they outperforms existing approaches on common NLP and vision tasks, however, only two methods are compared.
>
> The references provided by the reviewer are MoE-based methods for multi-task learning, and therefore are inapplicable in our setting. Our setting assumes a pre-trained dense model checkpoint, and MoEfication is the only MoE-based method fulfilling this condition that we are aware of.
>
> To strengthen the evaluation of our method, we implement dynamic inference method A-ViT [1] and include it as an additional baseline. A-ViT is spatially-adaptive in terms of its computational effort spent on an input image, similarly to our method (Figure 5a). Moreover, it also starts with a pre-trained dense model checkpoint, and thus is applicable in our setting. We provide the results of the ViT experiment updated with A-ViT baseline in Figure 2 in the rebuttal PDF. Our method outperforms the A-ViT baseline.
>
> > Furthermore, why does the authors does not provide any performance number in the table?
>
> We had considered plots as a more informative way to show the performance of our method. However, as per the reviewer’s suggestion, we have added them to the appendix of our current revision.
>
> ### Questions
>
> > Training Overhead:  (...)
>
> We provide those numbers in multiple places in the paper (e.g. L101-102; L193-194; Appendix H). Overall, this overhead is less than 1% of the overall training cost.
>
> > Adversarial Robustness:  (...)
>
> The adversarial robustness of dynamic inference methods is an issue that warrants an in-depth investigation in the form of a separate paper, as in Hong et al. [2]. We consider it too complex to fit within the scope of our work, but it’s an interesting and understudied future work direction.
>
> > Comparison with Pruning Techniques:  (...)
>
> D2DMoE is not a pruning or compression method, and therefore we do not consider such methods as suitable for direct comparison. We rather see model compression techniques as a complementary approach that can be applied alongside our method (see Section 5.6).
>
> > Scalability: (...)
>
> We do not see any specific issues that would prevent the application of our method to larger models, aside from the limited computational resources that prevent us from running such experiments.
>
> > The paper does not sufficiently address potential challenges in real-world deployment, such as latency variations, hardware compatibility, and integration with existing systems.
>
> > Real-world Deployment: (...)
>
> We did not list deployment limitations as they are not specific to our method but are general across dynamic inference methods. However, we followed the reviewer's suggestion and listed them in our paper to make sure that the reader understands them:
> 1. In contrast to static models, latency depends on the sample. In our case, it is upper-bounded by the latency of executing the forward pass with every token being routed to every expert.
> 2. The software has to support dynamic computational graphs.
> 3. Hardware compatibility is dependent on the method. We provide the wall-clock time measurement results for our implementation in the rebuttal PDF.
>
> We would like to thank the reviewer again for his time spent on reviewing our work. We hope that our answers resolve the reviewer’s concerns, and we are open to further discussion in case of any further questions. We also kindly ask the reviewer to reassess our work and adjust the score accordingly after taking into consideration our answers and the additional experiments from the response PDF.
>
> ##### References
>
> [1] Yin, Hongxu, et al. "A-vit: Adaptive tokens for efficient vision transformer." Proceedings of the IEEE/CVF conference on computer vision and pattern recognition. 2022.
>
> [2] Hong, Sanghyun, et al. "A panda? no, it's a sloth: Slowdown attacks on adaptive multi-exit neural network inference." arXiv preprint arXiv:2010.02432 (2020).

---

> > ### Comment · Reviewer_85BP · 2024-08-13
> >
> > I thank the reivewer's time and effort to give me a detailed response. However, I still think the dynamic-k routing is a slight modification of MOE's top-k routering.  Furthuremore, from fig. 4, we cannot observe the performance directly. Using loss value to represent the performance is not a good choice.  Although [1,2] are multi-task methods, they all provide pretrained models. The authors should attempt to compare them because the proposed method is quite similar to [1,2].

---

> ### Author Response · Authors · 2024-08-14
>
> We thank the reviewer for the response. While the brevity of their comment suggests that they may have already formed a definitive opinion about our work, we believe it is important to clarify our view to other participants. Apart from addressing the raised concerns, we also want to highlight how they have been presented. In particular, we find it challenging to understand why the **reviewer's response took so long, given that it is brief, somewhat vague, and largely reiterates previously stated points**. We point out other issues in our responses below. **We respectfully ask other participants to carefully consider both the reviewer's comments and our discussion to arrive at their own conclusions regarding the validity of the raised concerns.**
> > I still think the dynamic-k routing is a slight modification of MOE's top-k routering.
>
> **The reviewer seems to disregard our discussion, where we clearly state that dynamic-k is not our only contribution.** We also emphasize that reviewer uQnr praised the novelty of our router training scheme.
>
> We have provided a clear justification for the introduction of dynamic-k. This modification is evidently not that obvious given that dynamic-k was not adopted in earlier works. We demonstrate that static top-k gating is inappropriate for MoEs converted from dense models (Figure 2b) and that dynamic-k **enables consistent improvements** (Figure 4, Figure 5a).
> We also highlight that many influential ideas, such as the GELU activation function [1], mixup augmentation [2], dropout [3], or expert choice routing for MoEs [4] could be seen as __slight modifications__. Straightforward, yet well-motivated methods that provide consistent improvements are crucial for advancing the field, and simplicity often facilitates broader adoption.
>
> > Furthuremore, from fig. 4, we cannot observe the performance directly. Using loss value to represent the performance is not a good choice.
>
> **The reviewer initially described the cost savings shown in Fig. 4 as minimal. However, following our rebuttal, the reviewer apparently no longer seems to contest the improvement shown in the figure.** Instead, they raise a new concern shortly before the conclusion of the discussion period.
>
> We emphasize that the **reviewer’s statement is inaccurate, as we do report accuracy for classification tasks**. We use test loss only for LLM evaluation, as done in multiple other works. In particular, a significant portion of the **industry and research community heavily invests in research on scaling laws [5, 6]. These studies focus on improving model loss**, which indicates that it is widely considered as an important indicator of model performance.
>
> To further support our claim, **we provide a downstream evaluation of our Gemma models on the BoolQ dataset**. We take the base model, which achieves 68.40% accuracy, and convert it to MoE with D2DMoE and MoEfication and perform zero-shot evaluation. In the table below, we report **relative accuracy** of the models at different compute budgets. Similarly to the loss in Figure 4d, **our method largely retains the performance** across multiple compute budgets, while the **performance of MoEfication decreases significantly**.
>
> | Compute budget | 100%    | 90%    | 80%    | 70%    | 60%    | 50%    | 25%    | 10%    |
> |----------------|---------|--------|--------|--------|--------|--------|--------|--------|
> | D2DMoE         | 100.00% | 99.68% | 99.37% | 98.69% | 97.60% | 94.34% | 92.75% | 90.89% |
> | MoEfication    | 100.00% | 92.24% | 92.19% | 92.15% | 88.79% | 75.40% | 86.70% | 77.53% |
>
> > Although [1,2] are multi-task methods, they all provide pretrained models. The authors should attempt to compare them because the proposed method is quite similar to [1,2].
>
> We still consider the fact that the referenced works are multi-task learning solutions as a fundamental issue that renders any fair comparison impossible. The reviewer's response and his original review **fail to clarify how the referenced works are similar or relevant to our work**, with the only superficial connection being that they also use the MoE architecture. [7] targets Taskonomy and PASCAL-Context datasets, and [8] targets NYUD-v2 and PASCAL-Context datasets (all of them exclusively multi-task datasets). Neither of these works includes evaluations on baselines that are even remotely similar to ours, nor does our method target multi-task learning. Even though [7,8] might provide pre-trained models, comparison of models trained on different datasets **would mostly reflect performance differences stemming from the use of different training data** instead of those inherent to the method.
>
> We include the references in a separate response.

---

> ### Author Response · Authors · 2024-08-14
>
> #### References
> [1] Hendrycks, Dan, and Kevin Gimpel. "Gaussian error linear units (gelus)." arXiv preprint arXiv:1606.08415 (2016).
>
> [2] Zhang, Hongyi, et al. "mixup: Beyond empirical risk minimization." arXiv preprint arXiv:1710.09412 (2017).
>
> [3] Srivastava, Nitish, et al. "Dropout: a simple way to prevent neural networks from overfitting." The journal of machine learning research 15.1 (2014): 1929-1958.
>
> [4] Zhou, Yanqi, et al. "Mixture-of-experts with expert choice routing." Advances in Neural Information Processing Systems 35 (2022): 7103-7114.
>
> [5] Kaplan, Jared, et al. "Scaling laws for neural language models." arXiv preprint arXiv:2001.08361 (2020).
>
> [6] Clark, Aidan, et al. "Unified scaling laws for routed language models." International conference on machine learning. PMLR, 2022.
>
> [7] Chen, Zitian, et al. "Mod-squad: Designing mixtures of experts as modular multi-task learners." Proceedings of the IEEE/CVF Conference on Computer Vision and Pattern Recognition. 2023.
>
> [8] Fan, Zhiwen, et al. "M^3vit: Mixture-of-experts vision transformer for efficient multi-task learning with model-accelerator co-design." Advances in Neural Information Processing Systems 35 (2022): 28441-28457.

---

> ### Author Response · Authors · 2024-08-14
> **Further clarification on differences between D2DMoE and multi-task methods provided by the reviewer**
>
> To further emphasize the key differences between multi-task methods referenced in [7,8] and our work, we outline them below:
>
> * Our goal is to reduce inference cost by skipping the redundant computation during the model inference through the MoE framework. In contrast, the referenced works employ MoE primarily to solve the multi-task problem, with **efficiency being only a secondary concern**.
>
> * Both referenced methods use a classic formulation of MoE adapted for the multi-task setting (e.g. [7] optimizes mutual information between experts and tasks). **None of these works mentions activation sparsity or anything similar to our router training scheme**.
>
> * The cited works do always spend the **same amount of compute for each sample** (classic formulation of MoE), with the differences being present only between tasks for [7]. **In contrast, our dynamic-k gating adjusts the computational effort of the model on a per-sample and per-token basis**.
>
> * Our method allows for the **adjustment of computational cost after training** and during deployment through the change of the $tau$ hyperparameter. In contrast, [7,8] must retrain the model each time to target a different budget. While this may be not initially obvious from the FLOPs vs accuracy plots in ours and referenced works, we consider this a vital property of dynamic inference methods.
>
> * Finally, **we target scenarios with limited computational resources** for the adaptation of pre-trained model, while the references provided by the reviewer require a **significant compute budget to train models from scratch** (e.g. [7] reports training for 80 hours with 240 NVIDIA V100 GPUs).

---

### Official Review · Reviewer_v3M2 · 2024-07-23

**Soundness:** 2
**Presentation:** 4
**Contribution:** 3
**Rating:** 6
**Confidence:** 5

**Summary:**

The paper presents Dense to Dynamic-k Mixture-of-Experts (D2DMoE), a method to convert a dense model into a MoE model, that exploits the fact that activations in Transformer models are typically very sparse. The sparsity can be further improved by the addition of square Hoyer regularization during a light fine-tuning phase. Since the sparsity pattern changes over examples, the use of a dynamic selection rule is preferred. The strategy used to convert the dense model into a MLP is almost identical to that of Zhang et al. from 2022, but the loss function used to train the router is different. The router in this paper is trained to predict the norm of the output of the corresponding expert (and the experts with the an expected l2 norms over a certain dynamic threshold are selected). Finally, the present work also MoEfies not only the FFN layers as usual, but also all the linear layers in the MHA (for that, each linear layer is converted into a MLP of the same cost, minimizing the l2 error, similar to distillation).
The proposed approach is evaluated on different architectures (ViT, BERT, GPT2, Gemma) and benchmarks (including both vision and language). In all cases the proposed approach offeres a better FLOP-quality trade-off than that of Zhang et al. (2022), and achieves a quality similar to the original dense baseline but at significantly fewer FLOPs.

**Strengths:**

- The paper is very well structured and written. All the steps involved in the proposed method are clearly explained in the main paper, or briefly explained first and then thoroughly documented in the appendix.
- Figure 5a shows that the step done to improve sparsification is clearly useful. Although this doesn't show how the quality of the resulting model is affected, figure 4 shows that the quality of the original dense model is matched, or very nearly matched, with significantly fewer FLOPs (and better FLOP-quality trade-off than the MoEfication proposed by Zhang et al. in 2022).
- Figure 5b shows that more FLOPs (i.e. experts) are allocated to image regions with higher semantical meaning, as one intuitively would expect. This might not be a strength if one only cares about FLOP vs quality, but it certainly makes the approach more interpretable.

**Weaknesses:**

- The use of MoEs from scratch is criticized because the training difficulties that these have. However, the training recipes for these have improved significantly in the past two or three years, and are now a component of many state-of-the-art models (e.g. [Mixtral](https://mistral.ai/news/mixtral-of-experts/), [DeepSeekMoE](https://arxiv.org/abs/2401.06066), [Gemini 1.5](https://storage.googleapis.com/deepmind-media/gemini/gemini_v1_5_report.pdf)). Thus, I believe that these should definitely be considered as a baseline, otherwise the potential impact of this work is more limited.
- The proposed method involves many steps. They are well documented (see strengths), but makes the whole approach a bit cumbersome, in my opinion. Alternativelly, I wonder how the dynamic routing, the sparsity-indicuing regularization and other of the components used would perform when training a model from scratch.
- As usual, one must be carefull when doing only FLOP vs quality comparisons, since the proposed method, especially the dynamic selection of experts, may be hard to efficiently implement on modern hardware (GPUs, TPUs). I couldn't find this discussed anywhere in the paper, so I would appreciate more clarity on this (perhaps even additional plots showing Runtime vs quality, even if it's only on the appendix).

**Questions:**

See the comments mentioned in the weaknesses.

**Limitations:**

No particular ethical considerations affecting this work, in my opinion.

---

> ### Author Rebuttal · Authors · 2024-08-06
>
> We thank the reviewer for assessing our work and recognizing that our paper is well-written. Below, we address the reviewer's concerns.
>
> > The use of MoEs from scratch is criticized because the training difficulties that these have. However, the training recipes for these have improved significantly in the past two or three years, and are now a component of many state-of-the-art models (e.g. Mixtral, DeepSeekMoE, Gemini 1.5). Thus, I believe that these should definitely be considered as a baseline, otherwise the potential impact of this work is more limited.
>
> MoE models trained from-scratch may at first seem like an appropriate baseline due to similar architecture. However, note that the primary goal of training MoE from-scratch is to scale the parameter count of the model without affecting its *training* cost. In contrast, our aim is to reduce the *inference* cost of existing pre-trained models by skipping redundant computations while preserving the model size. All our baselines start from the same pre-trained model checkpoint, and we assume only a small budget for model adaptation, so we cannot conduct a fair comparison between our method and models suggested by the reviewer.
>
> > The proposed method involves many steps. They are well, documented (see strengths), but makes the whole approach a bit cumbersome, in my opinion. Alternativelly, I wounder how the dynamic routing, the sparsity-indicuing regularization and other of the components used would perform when training a model from scratch.
>
> While our method is multi-step, we show the robustness to the change of hyperparameters in Figures 6b, 6d, and 11. We agree that applying our method to training MoE from-scratch is an interesting future work direction. Expert contribution routing could be useful in augmenting the training of gating networks, which often perform similarly to fixed mappings to experts [1]. Moreover, as activation sparsity levels change as training progresses [2], one might investigate the application of dynamic-k gating for faster training. According to our knowledge, nothing similar to our contributions appeared in the context of MoE literature before.
>
> > As usual, one must be carefull when doing only FLOP vs quality comparisons, since the proposed method, especially the dynamic selection of experts, may be hard to efficiently implement on modern hardware (GPUs, TPUs). I couldn't find this discussed anywhere in the paper, so I would appreciate more clarity on this (perhaps even additional plots showing Runtime vs quality, even if it's only on the appendix).
>
> We address this concern in the joint response to all reviewers and in the rebuttal PDF in Figures 1 and 2, where we show our wall-clock time measurement results.
>
> We would like to thank the reviewer again for his time spent on reviewing our work. We hope that our answers resolve the reviewer’s concerns, and we are open to further discussion in case of any further questions. We also kindly ask the reviewer to reassess our work and adjust the score accordingly after taking into consideration our answers and the additional experiments from the response PDF.
>
> ##### References
>
> [1] Roller, Stephen, Sainbayar Sukhbaatar, and Jason Weston. "Hash layers for large sparse models." Advances in Neural Information Processing Systems 34 (2021): 17555-17566.
>
> [2] Wild, Cody, and Jesper Anderson. "Uncovering Layer-Dependent Activation Sparsity Patterns in ReLU Transformers." arXiv preprint arXiv:2407.07848 (2024).

---

> > ### Comment · Reviewer_v3M2 · 2024-08-13
> >
> > Thank you very much for addressing my concern, as well as those from other reviewers.
> >
> > I am still reluctant to increase my score, since I think that the method is quite cumbersome (as I mentioned) given the goal and what it achieves. For instance, the authors have mentioned during the rebuttal that the comparison with MoEs trained from-scratch is not fair because the goal of this work is "to reduce the inference cost of existing pre-trained models". If so, a really imporant and very popular strategy is missing as a baseline: distillation to a smaller model (dense or MoE).

---

> > > ### Author Response · Authors · 2024-08-14
> > >
> > > We thank the reviewer for the response. We understand the reviewer's concerns with distillation as a baseline, but - similarly to pruning - we see it more as a complementary model acceleration technique [1]. However, due to time constraints, we cannot provide any experiments that could demonstrate it to the reviewer and will only be able to include them in the camera-ready version.
> > >
> > > Additionally, we would like to point out that the CoFi method [2] evaluated in combination with our method in Section 5.6 does use distillation as a part of its training scheme. Our method still allows for computational saving when applied on top of CoFi. Input-dependent activation sparsity is an inherent property in almost all models, and therefore our method should be universally applicable.
> > >
> > > Finally, while we appreciate the thoroughness of the reviewer, to the best of our knowledge so far no papers have examined the compatibility of dynamic inference with all three major model compression methods—pruning, quantization, and distillation. In this context, our comparisons with pruning and quantization are already particularly comprehensive.
> > >
> > > We thank the reviewer for the discussion and his suggestions that helped us improve the paper.
> > >
> > > #### References
> > >
> > > [1] Han, Yizeng, et al. "Dynamic neural networks: A survey." IEEE Transactions on Pattern Analysis and Machine Intelligence 44.11 (2021): 7436-7456.
> > >
> > > [2] Xia, Mengzhou, Zexuan Zhong, and Danqi Chen. "Structured pruning learns compact and accurate models." arXiv preprint arXiv:2204.00408 (2022).

---

### Official Review · Reviewer_R1eJ · 2024-07-23

**Soundness:** 3
**Presentation:** 3
**Contribution:** 3
**Rating:** 6
**Confidence:** 3

**Summary:**

This paper proposes a method to convert a dense pre-trained transformer network into its sparse counterpart. The approach begins by fine-tuning a pre-trained network to sparsify its activation values. Subsequently, the method clusters the neurons in the MLP layers to form distinct experts and introduces a router network. Finally, the router network is fine-tuned to predict the norm of the output of each expert. Additionally, this work explores the possibility of modifying attention layers and clustering MLP layers using the GLU activation function. The results demonstrate that the proposed method is robust to high sparsity and outperforms baseline model

**Strengths:**

1. The computational cost of large models is a significant research problem. This work proposes a promising method that can reduce FLOPs by up to 60% without compromising performance.
2. This study conducts extensive experiments across different modalities, models, and datasets, making the results convincing.

**Weaknesses:**

1. This method introduces several hyperparameters during training and inference, which could potentially make it difficult to reproduce and deploy.
2. I find that Figure 2 does not justify dynamic-k very well, as most layers have a similar percentage of non-zero activations. However, this concern is alleviated after reviewing Figures 12 through 16.
3. Although this work sparsifies at a coarse granularity, which is beneficial for reducing latency, reporting wall-time would be more convincing than solely reporting FLOPs.

**Questions:**

After converting the attention weight matrices, is any router network introduced to determine which head is activated? In the main paper, I only read that the attention matrices are replaced with two-layer networks with comparable parameters and cost. I didn't find any rationale provided for this conversion.

**Limitations:**

1. Currently, this method is only evaluated on small networks, so it is unknown whether the proposed method can be effectively applied to larger networks.
2. Although this method leverages sparsity in activations to accelerate inference, it does not reduce the number of parameters, and may even increase them, unlike other methods such as pruning that simultaneously reduce the parameter count.

---

> ### Author Rebuttal · Authors · 2024-08-06
>
> We thank the reviewer for the time and effort spent on our paper. We are pleased to see that the reviewer recognizes the significance of our work and the robustness of our method across different experimental settings. Below, we answer to the issues raised by the reviewer:
>
> ### Weaknesses
>
> > This method introduces several hyperparameters during training and inference, which could potentially make it difficult to reproduce and deploy.
>
> While our method introduces several hyperparameters, we show that its performance is quite robust to hyperparameter selection (please see Figures 6b, 6d, and 11 for analysis of the impact of different hyperparameters). We found out that the staged nature of the conversion process significantly simplifies the tuning for both D2DMoE and MoEfication, as it allows us to tune and verify the performance of a stage before proceeding to the next one.
>
> > I find that Figure 2 does not justify dynamic-k very well, as most layers have a similar percentage of non-zero activations. However, this concern is alleviated after reviewing Figures 12 through 16.
>
> We want to clarify that the *average* (aggregated over all tokens from the entire test set) percentage of non-zero activations does not need to be different between layers for the use of dynamic-k gating to be justified. For a single FFN layer, we are interested in the variance of the number of non-zero activations. We show that this variance is significant (error bars in Figure 2b, top; we also plot the coefficient of variation in Figure 2b, bottom) for every FFN layer, hence executing the same number of experts for each input would be suboptimal.
>
> Note that some layers do exhibit different numbers of average non-zero activations (Figure 2b, top). While one could tune $k$ for each layer separately, it would be a cumbersome process, and to our best knowledge, there are no works in that direction. Our dynamic-k gating allows for an uneven spread of computation throughout the depth of the model without the need for tuning any hyperparameters (Figure 5a).
>
> > Although this work sparsifies at a coarse granularity, which is beneficial for reducing latency, reporting wall-time would be more convincing than solely reporting FLOPs.
>
> We address this concern in the joint response to all reviewers and in the rebuttal PDF in Figures 1 and 2, where we show our wall-clock time measurement results.
>
> ### Questions
>
> > After converting the attention weight matrices, is any router network introduced to determine which head is activated? In the main paper, I only read that the attention matrices are replaced with two-layer networks with comparable parameters and cost. I didn't find any rationale provided for this conversion.
>
> Please note that we do not introduce routing across the attention heads nor change how attention is calculated. We only replace the projection matrices (W_q, W_k, W_v, W_o). We introduce this modification to extend the computational savings from our method to attention layers without changing the self-attention mechanism itself.
>
> ### Limitations
>
> > Currently, this method is only evaluated on small networks, so it is unknown whether the proposed method can be effectively applied to larger networks.
>
> While our evaluation is done mostly on small-scale models, please note that we also demonstrate that our method works well with the 2B parameter Gemma model. Our computational budget does not allow for larger models, and as such we leave the exploration of this question for future work. However, since bigger models are more overparameterized and exhibit higher activation sparsity, we do not see any reasons why our method would not scale further. Finally, we see more improvement upon the baseline when D2DMoE is applied to the larger model (Gemma 2B) than when applied to smaller models; this suggests that our method scales better than the baseline (please see the discussion on why in Appendix C).
>
> > Although this method leverages sparsity in activations to accelerate inference, it does not reduce the number of parameters, and may even increase them, unlike other methods such as pruning that simultaneously reduce the parameter count.
>
> While this is true, the parameter overhead from our method is very low. We consider model compression methods such as pruning to be complementary to D2DMoE (see Section 5.6).
>
> We would like to thank the reviewer again for his time spent on reviewing our work. We hope that our answers resolve the reviewer’s concerns, and we are open to further discussion in case of any further questions. We also kindly ask the reviewer to reassess our work and adjust the score accordingly after taking into consideration our answers and the additional experiments from the response PDF.

---

> > ### Comment · Reviewer_R1eJ · 2024-08-14
> > **Response by Reviewer**
> >
> > Thank the authors for providing further clarification.
> >
> > I have read the rebuttal and other reviews. I decide to maintain my original score of 6.

---

> ### Author Response · Authors · 2024-08-14
>
> We thank the reviewer for the feedback and the discussion.

---

### Official Review · Reviewer_uQnr · 2024-07-25

**Soundness:** 4
**Presentation:** 3
**Contribution:** 4
**Rating:** 6
**Confidence:** 4

**Summary:**

This paper improves over MoEfication with four innovations: (i) enforcing higher activation sparsity; (ii) directly predict the norm of the output of each expert; (iii) dynamic-k expert selection scheme; and (iv) generalization to any standalone linear layer. The resulting method achieves significant improvements in terms of cost-vs-performance trade-offs in text classification, image classification, and language modeling.

**Strengths:**

- The proposed method is a nice addition of MoEfication, in terms of both performance and generalization (to any standalone linear layer). It makes this kind of method more usable in practice.

- I like the idea of directly predicting the norm so as to achieve dynamic-k expert selection. This dynamic scheme is novel and bridges the fields of dynamic inference and MoE.

- The experiments illustrate that the proposed method is applicable in various settings, including text classification, image classification, and language modeling.

**Weaknesses:**

- The baseline seems weak. ZTW only receives a few citations and is not published at a top machine learning conference/journal.

- Only FLOPs are reported and it is a poor indicator of practical latency/throughput, especially in the LLM era.

- It is not clear if the proposed method could be combined with other methods to show its broader applicability. In particular, quantization is a very general method and is almost adopted as a default in many practical applications.

- The writing could be improved. For example, more technical details of MoEfication should be included.

- In all figures, the lines are continuous. However, these lines should have been a result of interpolation. The original point values are missing.

**Questions:**

NA

**Limitations:**

Please see weakness.

---

> ### Author Rebuttal · Authors · 2024-08-06
>
> We thank the reviewer for the time and effort spent on our paper. We are pleased that the reviewer recognizes the novelty, applicability, and performance of our method. Below, we address the issues raised by the reviewer.
>
> > The baseline seems weak. ZTW only receives a few citations and is not published at a top machine learning conference/journal.
>
> ZTW [1] was published at NeurIPS 2021, and its extended version [2] was published in Neural Networks in 2023. We apologize for the confusion. We cited the extension believing it is more relevant due to being more recent, but in our new revision we cite both versions of that paper.
>
> > Only FLOPs are reported and it is a poor indicator of practical latency/throughput, especially in the LLM era.
>
> We address this concern in the joint response to all reviewers and in the rebuttal PDF in Figures 1 and 2, where we show our wall-clock time measurement results.
>
> > It is not clear if the proposed method could be combined with other methods to show its broader applicability. In particular, quantization is a very general method and is almost adopted as a default in many practical applications.
>
> Multiple studies such as [4,5,6,7,8] show that dynamic computation methods combine well with quantization and pruning. In Section 5.6, we demonstrate that our method integrates effectively with the existing structured pruning method CoFi [3]. For reviewers' convenience, we perform a similar experiment with 8-bit and 16-bit dynamic quantization and provide the results in the rebuttal PDF. Our method integrates seamlessly with quantization.
>
> > The writing could be improved. For example, more technical details of MoEfication should be included.
>
> Following the reviewer's suggestion, we have included a more detailed description of MoEfication in the paper.
>
> > In all figures, the lines are continuous. However, these lines should have been a result of interpolation. The original point values are missing.
>
> We changed the plots to add points as suggested by the reviewer.
>
> We would like to thank the reviewer again for his time spent on reviewing our work. We hope that our answers resolve the reviewer’s concerns, and we are open to further discussion in case of any further questions. We also kindly ask the reviewer to reassess our work and adjust the score accordingly after taking into consideration our answers and the additional experiments from the response PDF.
>
> ##### References
>
> [1] "Zero time waste: Recycling predictions in early exit neural networks.", Wołczyk et al., NeurIPS2021.
>
> [2] "Zero time waste in pre-trained early exit neural networks.", Wójcik et al, Neural Networks 168 (2023).
>
> [3] "Structured Pruning Learns Compact and Accurate Models.", Xia et al., ACL2022.
>
> [4] "Mixture of Quantized Experts (MoQE): Complementary Effect of Low-bit Quantization and Robustness.", Young, arXiv 2023.
>
> [5] "The Optimal BERT Surgeon: Scalable and Accurate Second-Order Pruning for Large Language Models.", Kurtic et al., EMNLP 2022.
>
> [6] “Deep Compression: Compressing Deep Neural Networks with Pruning, Trained Quantization and Huffman Coding”, Han et al., ICLR2016.
>
> [7] “McQueen: Mixed Precision Quantization of Early Exit Networks”, Saxena et al., BMVC2023.
>
> [8] “Two sparsities are better than one: unlocking the performance benefits of sparse–sparse networks.”, Hunter et al., Neuromorphic Computing and Engineering 2022 Volume 2 Number 3.

---

> > ### Comment · Reviewer_uQnr · 2024-08-07
> >
> > Thanks for the reply and the additional experiments. Most of my concerns are addressed except two:
> >
> > - Although the baseline is from a top conference, it seems too old. Additional experiments with newer baselines may not be necessary during this rebuttal period, but I highly recommend trying out more recent ones.
> >
> > - If your method does not hurt accuracy, and since quantization usually does not hurt accuracy too much, I'm not surprised that when combining your method with quantization, the accuracy will be maintained. However, this may not be true for latency. Therefore, it's better to report both accuracy and latency, when combining your method with quantization.

---

> ### Author Response · Authors · 2024-08-10
>
> We appreciate the reviewer’s involvement and are grateful for a very swift response.
>
> > Although the baseline is from a top conference, it seems too old. (...)
>
> The extension paper from November 2023 demonstrates that ZTW remains a SOTA early exit method and outperforms other baselines such as GPF [1] (ACL2021) and L2W [2] (ECCV2022).
>
> In the response to reviewer 85BP we have also added A-ViT [3] (CVPR2022), with the results available in Figure 3 of the rebuttal PDF.
>
> Overall, we compare our method against four baselines (ZTW, MoEfication[4], CoFi[5], A-ViT) and conduct experiments on four datasets from different modalities. This surpasses the evaluation scope of other established works on dynamic architectures published at top conferences, which often evaluate on fewer datasets and baselines or compare against static models instead ([3], CVPR2022; [4], ACL 2022; [5], CVPR2023; [6], EMNLP2023; [7], NeurIPS2021; [8], NeurIPS2022; [9] ICLR2023).
>
> > (...) it's better to report both accuracy and latency, when combining your method with quantization.
>
> To show that quantization reduces the latency of D2DMoE, we modify our kernels to handle `float16` and `int8` data types. We perform a similar experiment to the one from Figure 1 of the rebuttal PDF: 1) we sample gating decisions from the Bernoulli distribution with probability $p$ 2) measure the execution time of our experts for the three data type variants. We present the latency (ms) in the tables below:
>
> ### RTX 4090
> | p  	 | 0.0 	 | 0.1 	 | 0.2 	 | 0.3 	 | 0.4 	 | 0.5 	 | 0.6 	 | 0.7 	 | 0.8 	 | 0.9 	 | 1.0 	 |
> |---------|----------|----------|----------|----------|----------|----------|----------|----------|----------|----------|----------|
> | float32 | 0.005 | 0.009 | 0.013 | 0.018 | 0.023 | 0.028 | 0.033 | 0.038 | 0.042 | 0.047 | 0.052 |
> | float16 | 0.004 | 0.005 | 0.007 | 0.009 | 0.011 | 0.014 | 0.016 | 0.018 | 0.021 | 0.024 | 0.027 |
> | int8    | 0.004 | 0.004 | 0.005 | 0.007 | 0.008 | 0.009 | 0.010 | 0.011 | 0.012 | 0.013 | 0.014 |
>
> ### A100
> | p  	 | 0.0 	 | 0.1 	 | 0.2 	 | 0.3 	 | 0.4 	 | 0.5 	 | 0.6 	 | 0.7 	 | 0.8 	 | 0.9 	 | 1.0 	 |
> |---------|----------|----------|----------|----------|----------|----------|----------|----------|----------|----------|----------|
> | float32 | 0.006 | 0.009 | 0.012 | 0.015 | 0.019 | 0.022 | 0.025 | 0.028 | 0.031 | 0.035 | 0.038 |
> | float16 | 0.006 | 0.007  | 0.008 | 0.010 | 0.011 | 0.013 | 0.014 | 0.016 | 0.017 | 0.019 | 0.021 |
> | int8    | 0.007 | 0.008 | 0.009 | 0.010 | 0.011 | 0.012 | 0.014 | 0.015 | 0.016 | 0.017 | 0.019 |
>
> The results show that both the higher activation sparsity (lower $p$) of our method and lower-precision data types are complementary in terms of wall-clock time reduction. While we see a smaller improvement from using `int8` over `float16` on A100, we attribute this to differences between GPU architectures and software support for low-precision arithmetic.
>
> ##### References
> [1] Liao, Kaiyuan, et al. "A global past-future early exit method for accelerating inference of pre-trained language models." Proceedings of the 2021 conference of the north american chapter of the association for computational linguistics: Human language technologies. 2021.
>
> [2] Han, Yizeng, et al. "Learning to weight samples for dynamic early-exiting networks." European conference on computer vision. Cham: Springer Nature Switzerland, 2022.
>
> [3] Yin, Hongxu, et al. "A-vit: Adaptive tokens for efficient vision transformer." Proceedings of the IEEE/CVF conference on computer vision and pattern recognition. 2022.
>
> [4] Zhang, Zhengyan, et al. "MoEfication: Transformer Feed-forward Layers are Mixtures of Experts." Findings of the Association for Computational Linguistics: ACL 2022. 2022.
>
> [5] Chen, Xuanyao, et al. "Sparsevit: Revisiting activation sparsity for efficient high-resolution vision transformer." Proceedings of the IEEE/CVF Conference on Computer Vision and Pattern Recognition. 2023.
>
> [6] Tan, Shawn, et al. "Sparse Universal Transformer." Proceedings of the 2023 Conference on Empirical Methods in Natural Language Processing. 2023.
>
> [7] Rao, Yongming, et al. "Dynamicvit: Efficient vision transformers with dynamic token sparsification." Advances in neural information processing systems 34 (2021): 13937-13949.
>
> [8] Schuster, Tal, et al. "Confident adaptive language modeling." Advances in Neural Information Processing Systems 35 (2022): 17456-17472.
>
> [9] Chataoui, Joud, and Mark Coates. "Jointly-Learned Exit and Inference for a Dynamic Neural Network." The Twelfth International Conference on Learning Representations. 2023.

---

> > ### Comment · Reviewer_uQnr · 2024-08-13
> >
> > Thank you for conducting the additional experiments. I don’t see any major concerns that would result in a rejection of this paper. However, after reading the comments from the other reviewers, I think there is still room for improvement in experiments. As a result, I decide to maintain my current score.

---

> ### Author Response · Authors · 2024-08-14
>
> We appreciate the reviewer's response, active participation in the discussion, and support for our work. Their feedback has been valuable in helping us improve our paper.

---

### Official Review · Reviewer_VwY8 · 2024-07-30

**Soundness:** 3
**Presentation:** 3
**Contribution:** 3
**Rating:** 5
**Confidence:** 5

**Summary:**

The paper introduces a method called D2DMoE aimed at enhancing the efficiency of transformer models. D2DMoE implements a dynamic-k routing mechanism that allows the model to select a variable number of experts based on the input. The method leverages the inherent activation sparsity in transformer models to reduce the number of active parameters during inference, leading to significant computational savings—up to 60%—without compromising performance. The approach demonstrates that by converting dense layers into Mixture-of-Experts (MoE) layers, transformer models can achieve better accuracy-sparsity trade-offs, making them more efficient for various NLP and vision tasks.

**Strengths:**

The paper presents a thorough empirical evaluation across multiple tasks (image classification, text classification, language modeling) and model architectures (ViT, BERT, GPT-2, Gemma). The experiments compare against relevant baselines and demonstrate consistent improvements.

**Weaknesses:**

In my view, this paper appears to be primarily a repackaged sparse-activated pruning technique stemming from the MoE concept. Several concerns arise:

1. Limited comparison with alternative sparsification methods: The paper predominantly contrasts with MoEfication, neglecting a thorough analysis against other compression or sparsification strategies beyond early-exit techniques. Consequently, the comparative experiments presented lack depth and fail to be fully convincing.

2. Lack of practical acceleration results: The paper only presents theoretical reductions in FLOPs and parameter counts. Without actual inference acceleration results on real hardware (e.g., V100, H100, GTX-4090Ti), it's impossible to assess the practical benefits of the method. The additional overhead from the gating mechanism could potentially negate some of the theoretical gains.

3. Questionable novelty of Dynamic-k gating: The proposed expert selection based on ℓ2-norm of output is indeed more reminiscent of pruning techniques than traditional MoE approaches. This calls into question the novelty of the method when viewed in the context of pruning literature, and makes the comparisons with MoE methods potentially unfair or irrelevant.

4. Limited novelty and inadequate discussion of related work: Many of the proposed operations bear similarities to existing sparse activation pruning methods. The paper fails to adequately discuss these connections, instead focusing on less relevant work. This omission of crucial related work in pruning literature significantly undermines the claimed novelty of the approach.

5. Lack of discussion of some SOTA sparse methods: such as Pruner-Zero: Evolving Symbolic Pruning Metric From Scratch for Large Language Models. ICML2024.

6. Theoretical foundation: The paper lacks a strong theoretical justification for why the proposed methods work better, which becomes even more critical given the concerns about its relationship to existing pruning techniques.

**Questions:**

See weaknesses.

---

> ### Author Rebuttal · Authors · 2024-08-06
>
> We thank the reviewer for the time spent reviewing our work. We are grateful that the reviewer recognizes that our paper presents a thorough empirical evaluation and compares the proposed method against relevant baselines. Below, we present the responses to the issues listed by the reviewer.
>
> > ​​In my view, this paper appears to be primarily a repackaged sparse-activated pruning technique stemming from the MoE concept.
>
> > Limited comparison with alternative sparsification methods: (...)
>
> > Lack of discussion of some SOTA sparse methods: (...)
>
> We would like to highlight that our method is not a pruning technique, but a dynamic inference method. Dynamic inference methods (also known as conditional computation or adaptive inference) [8] are a distinct area of research fundamentally different from pruning approaches. We list the exact differences below:
> 1) Pruning focuses on model **weights**, while dynamic inference results in sparsity in model **activations**.
> 2) During inference, the pruned model **statically** uses the same subset of weights for all inputs. In contrast, dynamic inference methods **dynamically** select an appropriate subset of weights on a per-input basis instead of removing them.
> 3) Pruning is usually concerned with **model compression**, while dynamic inference mostly focuses on **inference acceleration**. The two objectives are not always correlated; in particular, unstructured pruning fails to provide any speedups on GPUs in practice due to the lack of proper hardware support for sparse matrix multiplications [1, 2].
>
> It has already been shown that combining weight sparsity and activation sparsity can lead to higher speedups than with weight sparsity alone [2]. Since conditional computation methods such as MoE can be seen as a form of structured activation sparsity [1], we consider our method and weight pruning as complementary.
>
> Crucially, we highlight that we do explore the relationship of our method to pruning in our paper. In Section 5.6 we perform an experiment that tests our method for compatibility with pruning, where we demonstrate that our method indeed improves the performance of a network pruned by the CoFi structured pruning method [9].
>
> > Questionable novelty of Dynamic-k gating: (...)
>
> > Limited novelty and inadequate discussion of related work: (...)
>
> We would like to gently disagree with the reviewer in regards to the lack of novelty of our method and point out that other reviewers did not raise similar concerns and even praised the novelty of our work (reviewer uQnr). To the best of our knowledge, we were the first to propose MoE routing based on the norms of the outputs of the experts. Our expert contribution routing is conceptually consistent, significantly improves performance (Figure 6a), and is not dependent on ReLU (Figure 6c). The use of the ℓ2-norm (or any norm) is widespread in the machine learning literature (e.g. adversarial examples [3], knowledge distillation [4], normalization [5]), and we do not see any connection to pruning in particular.
>
> Please also note that expert contribution routing and dynamic-k gating are separate contributions (Figure 6a). We show that static top-k gating is inappropriate for MoEs converted from dense models (Figure 2b), a setup first proposed by Zhang et al. [6]. Similarly, we are the first to show the existence of this problem, and we propose dynamic-k gating as a remedy - that in our view also has no connection to the pruning literature.
>
> Following the discussion with the reviewer, in addition to Section 5.6, we add the discussion on pruning and its relation to our method to the paper to better highlight the differences for the readers.
>
> > Theoretical foundation: (...)
>
> While we acknowledge the importance of strong theoretical foundations in machine learning, we focus on empirical evaluation as done in works similar to ours [2,6,7,8].
>
> > Lack of practical acceleration results: (...)
>
> We address this concern in the joint response to all reviewers and in the rebuttal PDF in Figures 1 and 2, where we show our wall-clock time measurement results.
>
> We would like to thank the reviewer again for his time spent on reviewing our work. We hope that our answers resolve the reviewer’s concerns, and we are open to further discussion in case of any further questions. We also kindly ask the reviewer to reassess our work and adjust the score accordingly, considering our answers and the additional rebuttal experiments.
>
> ##### References
> [1] Hoefler, Torsten, et al. "Sparsity in deep learning: Pruning and growth for efficient inference and training in neural networks." Journal of Machine Learning Research 22.241 (2021): 1-124.
>
> [2] Hunter, Kevin, Lawrence Spracklen, and Subutai Ahmad. "Two sparsities are better than one: unlocking the performance benefits of sparse–sparse networks." Neuromorphic Computing and Engineering 2.3 (2022): 034004.
>
> [3] Costa, Joana C., et al. "How deep learning sees the world: A survey on adversarial attacks & defenses." IEEE Access (2024).
>
> [4] Gou, Jianping, et al. "Knowledge distillation: A survey." International Journal of Computer Vision 129.6 (2021): 1789-1819.
>
> [5] Hoffer, Elad, et al. "Norm matters: efficient and accurate normalization schemes in deep networks." Advances in Neural Information Processing Systems 31 (2018).
>
> [6] Zhang, Zhengyan, et al. "Moefication: Transformer feed-forward layers are mixtures of experts." arXiv preprint arXiv:2110.01786 (2021).
>
> [7] Shazeer, Noam, et al. "Outrageously large neural networks: The sparsely-gated mixture-of-experts layer." arXiv preprint arXiv:1701.06538 (2017).
>
> [8] Han, Yizeng, et al. "Dynamic neural networks: A survey." IEEE Transactions on Pattern Analysis and Machine Intelligence 44.11 (2021): 7436-7456.
>
> [9] Xia, Mengzhou, Zexuan Zhong, and Danqi Chen. "Structured Pruning Learns Compact and Accurate Models." Proceedings of the 60th Annual Meeting of the Association for Computational Linguistics (Volume 1: Long Papers). 2022.

---

> > ### Comment · Reviewer_VwY8 · 2024-08-12
> > **About dynamic inference method**
> >
> > The authors highlight that their work is a dynamic inference approach, so are there any comparisons with previous works on dynamic pruning?
> >
> > There are also more advanced approaches to dynamic inference, such as dynamic token sparsity, etc. This work seems to have no advantage against these approaches

---

> ### Author Response · Authors · 2024-08-12
>
> We thank the reviewer for the comments. We understand we have addressed all the other concerns.
>
> >There are also more advanced approaches to dynamic inference, such as dynamic token sparsity, etc. This work seems to have no advantage against these approaches
>
> We refer the reviewer to Figure 3 of the rebuttal PDF, where we add A-ViT [1], a dynamic token sparsity method, as a baseline for the response to reviewer 85BP. The results show that we outperform A-ViT.
>
> >The authors highlight that their work is a dynamic inference approach, so are there any comparisons with previous works on dynamic pruning?
>
> According to a recent survey [2], at least two groups of works use the term “dynamic pruning”.
>
> Dynamic pruning approaches such as [3, 4, 5, 6] differ from static pruning because they allow the network structure to change by pruning or reactivating connections during training. The resulting model is static and therefore should complement dynamic inference methods such as ours (as shown in Section 5.6).
>
> A small subset of dynamic inference works do use the “dynamic pruning” term for methods that dynamically select channels for execution in convolutional layers [7, 8, 9]. However, they all target exclusively CNNs, while our work focuses on Transformer models.
>
> ##### References
> [1] Yin, Hongxu, et al. "A-vit: Adaptive tokens for efficient vision transformer." Proceedings of the IEEE/CVF conference on computer vision and pattern recognition. 2022.
>
> [2] He, Yang, and Lingao Xiao. "Structured pruning for deep convolutional neural networks: A survey." IEEE transactions on pattern analysis and machine intelligence (2023).
> [3] Lin, Tao, et al. "Dynamic Model Pruning with Feedback." ICLR-International Conference on Learning Representations. 2020.
>
> [4] Ruan, Xiaofeng, et al. "DPFPS: Dynamic and progressive filter pruning for compressing convolutional neural networks from scratch." Proceedings of the AAAI Conference on Artificial Intelligence. Vol. 35. No. 3. 2021.
>
> [5] Lin, Shaohui, et al. "Accelerating Convolutional Networks via Global & Dynamic Filter Pruning." IJCAI. Vol. 2. No. 7. 2018.
>
> [6] Chen, Zhiqiang, et al. "Dynamical channel pruning by conditional accuracy change for deep neural networks." IEEE transactions on neural networks and learning systems 32.2 (2020): 799-813.
>
> [7] Elkerdawy, Sara, et al. "Fire together wire together: A dynamic pruning approach with self-supervised mask prediction." Proceedings of the IEEE/CVF Conference on Computer Vision and Pattern Recognition. 2022.
>
> [8] Gao, Xitong, et al. "Dynamic Channel Pruning: Feature Boosting and Suppression." International Conference on Learning Representations.
>
> [9] Lin, Ji, et al. "Runtime neural pruning." Advances in neural information processing systems 30 (2017).

---

### Author Rebuttal · Authors · 2024-08-06

We thank all the reviewers for the time and effort spent on our work and their valuable comments that helped us improve our work. We appreciate that our work has been praised by the reviewers for its thorough empirical evaluation (reviewers VwY8, R1eJ), novelty (reviewer uQnr), the generality of our method (reviewer uQnr), good empirical results (reviewers VwY8, uQnr, and R1eJ), and the paper being well-written (v3M2).

## Rebuttal summary
We highlight the main points that changed in our new revision thanks to the valuable feedback that we received from the reviewers:
- Reviewers VwY8, uQnr, R1eJ, and v3M2 requested wall-clock time measurements of the performance of our method. To strengthen our work we provide an efficient implementation of D2DMoE and show its performance in the rebuttal PDF. In Figure 1, we show that FLOPs for D2DMoE correspond to actual latency and that our method can significantly speed up the inference (63% reduction of wall-clock time with a negligible performance drop). Moreover, in Figure 2 we show end-to-end accuracy vs latency plots that show D2DMoE can reduce the end-to-end computation by around 25% time without any performance drop. We consider our efficient implementation of D2DMoE as a strong technical contribution of our work.
- Reviewer 85BP requested additional baselines. As a result, we add A-ViT [1] into our comparison. A-ViT is a dynamic inference method that saves compute by dropping tokens. The results are presented in Figure 3 of the rebuttal PDF. D2DMoE outperforms this baseline as well.
- Reviewer uQnr asked about the integration of D2DMoE with quantization. Therefore, in Figure 4 of the rebuttal PDF, we provide an additional experiment with quantization. We show that our method integrates seamlessly with 16-bit and 8-bit quantization, further proving the robustness of our method.

Below, we elaborate on wall-clock measurements and our rationale for using FLOPs in more detail.

## Wall-clock time measurement results

We implement the forward pass of our MoE module using GPU kernels written in Triton [2] and employ several optimizations for our implementation, including an efficient memory access pattern, kernel fusion, and configuration auto-tuning. As suggested by Tan et al. [4], our implementation also avoids unnecessary copies when grouping tokens.

We verify the performance of our implementation for a single FFN MoE layer in isolation by feeding it with synthetic random data. We measure the execution time of our layer for a range of different numbers of experts executed on average and compare it to the corresponding MLP module. The results, presented in Figure 1 of the rebuttal PDF, show that our implementation has almost no overhead, and that FLOPs for D2DMoE perfectly correlate with the wall-clock time of execution.

Furthermore, we want to ensure that these gains translate to real-world usage speedups. We therefore use our new implementation in a ViT-B-converted D2DMoE model and measure the average sample processing time on the entire ImageNet-1k test set. As suggested by reviewer v3M2, in Figure 2 of the rebuttal PDF we plot the accuracy vs wall-clock time trade-off of our method. While a small overhead when compared to the ViT-B baseline is visible, our method still allows for significant savings. This shows the speed-up potential of D2DMoE in practice.

We have added the above wall-clock measurement experiments to our paper, along with a detailed description of our Triton implementation and its source code.

## FLOPs rationale
Finally, we provide our reasoning for using FLOPs as the main metric in our work instead of using wall-clock time measurements:
1. Wall-clock measurements are heavily affected by the choice of device hardware. Even if the device model (i.e. GPU model) is the same, the environment (cooling efficiency, temperature, package versions, I/O load by other users) may be different and can significantly affect the result. As such, wall-clock time does not allow for reliable comparisons to results presented in other works.
2. Implementing efficient GPU kernels is often non-trivial, and entire papers have been published with their sole contribution being an efficient implementation of MoE [2, 3]. Latencies obtained with poorly implemented and slow kernels could suggest to the reader that the method is GPU-unfriendly, even if it is a problem of only that specific implementation. We emphasize the rapid progress in research on efficient MoE implementations, exemplified by the recent work of Tan et al. [4], which demonstrates significant advancements compared to Gale et al. [3].
3. Finally, as discussed in "The hardware lottery" paper by Hooker, S. [5], compatibility with the current hardware should not overshadow the ideas proposed in research papers. FLOPs, as a metric for computational complexity, are independent of hardware choice and can be used to compare algorithms rather than their specific implementations.

We thank the reviewers for their valuable insights. We hope that our response and additional experiments fully address the concerns of the reviewers, and we are open to further discussion should reviewers have any additional inquiries.

##### References
[1] Yin, Hongxu, et al. "A-vit: Adaptive tokens for efficient vision transformer." Proceedings of the IEEE/CVF conference on computer vision and pattern recognition. 2022.

[2] Tillet, Philippe, Hsiang-Tsung Kung, and David Cox. "Triton: an intermediate language and compiler for tiled neural network computations." Proceedings of the 3rd ACM SIGPLAN International Workshop on Machine Learning and Programming Languages. 2019.

[3] Gale, Trevor, et al. "Megablocks: Efficient sparse training with mixture-
of-experts." Proceedings of Machine Learning and Systems 5 (2023).

[4] Tan, Shawn, et al. "Scattered Mixture-of-Experts Implementation." arXiv preprint arXiv:2403.08245 (2024).

[5] Hooker, Sara. "The hardware lottery." Communications of the ACM 64.12 (2021): 58-65.

---

### Author Response · Authors · 2024-08-14

We thank all the reviewers for the discussion.

We have addressed each reviewer's concerns in detail, including the additional experiments and clarifications provided in our initial rebuttal regarding FLOPs vs latency, the A-ViT baseline, and quantization. During the discussion period, we also presented new results with quantization for reviewer uQnr. We also added downstream evaluations for reviewer 85BP, despite receiving their feedback just a day before the end of the discussion period.

Our general impression was that the reviewers’ main concerns revolved around the latency of our method and its compatibility with other model acceleration techniques. We have provided an efficient GPU kernel for our method, which we consider a significant technical contribution. We demonstrated that our method provides a ​​63% wall-clock time reduction for FFN modules and around 30% reduction when evaluated end-to-end. We believe that we have adequately addressed the above-mentioned issues through the discussion.

We are pleased to note that the three reviewers who recommended acceptance remain positive about our work after the discussion and were satisfied with the responses we provided. We would like to express our gratitude to all the reviewers for their engagement during the discussion period and constructive feedback that helped us improve our paper.

---

### Decision · Program_Chairs · 2024-09-25

**Decision:**

Accept (poster)

**Comment:**

The submission proposes a dynamic-k routing mechanism to select the number of experts in a MoE setting.  The submission initially received split reviews, which improved during the rebuttal process.  All but one reviewer now recommend acceptance.  The remaining reviewer has concerns about novelty, noting other papers that have used a dynamic-k routing mechanism in some way.  The submission also provides an interesting practical implementation and benchmarking results that appear to be of value in themselves.  The authors are strongly recommended to include citations to and discussion about the references pointed out by Reviewer 85BP - this will strengthen the correctness and relevance of the paper.